

# The *g*-function and defect changing operators from wavefunction overlap on a fuzzy sphere

Zheng Zhou[1,2], Davide Gaiotto[1], Yin-Chen He[1] and Yijian Zou[1]

**1** Perimeter Institute for Theoretical Physics, Waterloo, Ontario, Canada N2L 2Y5
**2** Department of Physics and Astronomy, University of Waterloo,
Waterloo, Ontario, Canada N2L 3G1

## Abstract

Defects are common in physical systems with boundaries, impurities or extensive measurements. The interaction between bulk and defect can lead to rich physical phenomena. Defects in gapless phases of matter with conformal symmetry usually flow to a defect conformal field theory (dCFT). Understanding the universal properties of dCFTs is a challenging task. In this paper, we propose a computational strategy applicable to a line defect in arbitrary dimensions. Our main assumption is that the defect has a UV description in terms of a local modification of the Hamiltonian so that we can compute the overlap between low-energy eigenstates of a system with or without the defect insertion. We argue that these overlaps contain a wealth of conformal data, including the *g*-function, which is an RG monotonic quantity that distinguishes different dCFTs, the scaling dimensions of defect creation operators $\Delta_\alpha^{+0}$ and changing operators $\Delta_\alpha^{+-}$ that live on the intersection of different types of line defects, and various OPE coefficients. We apply this method to the fuzzy sphere regularization of 3D CFTs and study the magnetic line defect of the 3D Ising CFT. Using exact diagonalization and DMRG, we report the non-perturbative results $g = 0.602(2), \Delta_0^{+0} = 0.108(5)$ and $\Delta_0^{+-} = 0.84(5)$ for the first time. We also obtain other OPE coefficients and scaling dimensions. Our results have significant physical implications. For example, they constrain the possible occurrence of spontaneous symmetry breaking at line defects of the 3D Ising CFT. Our method can be potentially applied to various other dCFTs, such as plane defects and Wilson lines in gauge theories.



# 1   Introduction

Defects occur naturally in many physical systems. In general, a $p$-dimensional "defect" is a modification of a $d$-dimensional system that is localized in $d-p$ directions. If the original system is endowed with translation symmetry, we can consider defects that preserve $p$ translation generators. Boundaries and interfaces are natural examples of $(d-1)$-dimensional defects. Impurities are also a classical source of defects with interesting dynamics, like in the Kondo problem [1, 2]. Other defects may occur naturally as a low-energy description of some very massive excitation.[1] More recently, it has been shown that defects appear in the description of certain quantum measurements or decoherence on a quantum many-body state [3–9]. In a continuum or low-energy limit, the notion of defect is as primitive and universal as the notion of quantum field theory itself. In the same manner as a given quantum field theory capturing a whole universality class of systems, local defects in a quantum field theory capture universality classes of local modifications of these systems. Defects have played a significant role in the development of modern theoretical physics. For example, one of the first successful applications of the renormalization group is the Kondo problem [1]. Defects are also useful to understand the underlying mathematical structure of quantum field theories, such as symmetries [10] and anomalies [11]. In systems with topological order, anyons can also be viewed as topological line defects. Last but not least, the confinement of gauge theories concerns the properties of a special type of defect–the Wilson loop [12, 13].

The renormalization group flow acts on defects as well as the bulk system. For gapless phases of matter that possess an effective conformal field theory (CFT) description at low energies, defects, and boundaries often flow to conformal fixed points described by "defect CFTs" (dCFTs), also known as "conformal defects" [14–21]. See also [22–47] for a non-exhaustive overview of theoretical aspects of conformal defects.

---

[1]For example, the low-energy description of massive excitations in gapped systems with topological order involves anyons, formalized as topological line defects in a topological quantum field theory. In electromagnetism, a Wilson line provides the low-energy description of a massive charged particle. At a first-order phase transition, the properties of a domain wall between two phases may be encoded into an interface between the two low-energy theories describing the phases. In all of these examples, the low-energy description of the massive excitation will also include extra degrees of freedom such as the positions of the excitations, which couple to local operators on the defect to give a low-energy effective action.

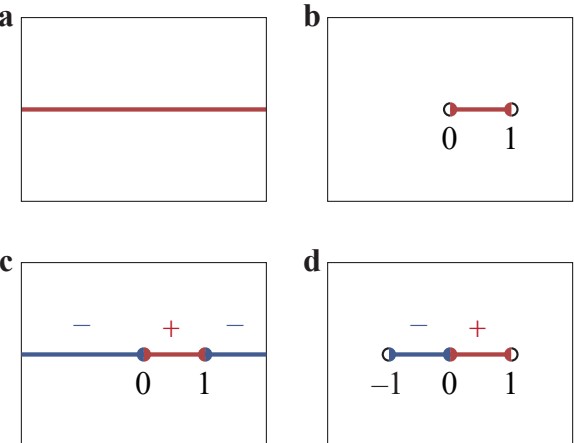

Figure 1: An illustration of the path integral configurations of (a) an infinite long defect; (b) the two-point function of defect creation operator; (c) the two-point function of defect changing operator; (d) the three-point function of defect creation and changing operators.

In the following, we will focus on one-dimensional defects, aka line defects. To formulate a CFT with a line defect, we start with a CFT in $d \geq 2$ dimensional Euclidean spacetime with action $S_{\text{CFT}}$.[2] The CFT has an infinite set of (global) conformal primary operators $\phi_\alpha$. Each primary operator $\phi_\alpha$ is associated to a scaling dimension $\Delta_\alpha$ and $SO(d)$ spin $s_\alpha$. Primary operators together with their derivatives generate irreducible representations of the conformal group $SO(d + 1, 1)$. A simple line defect in the CFT can be obtained by turning on a relevant perturbation on a line $x^1 = x^2 = \cdots x^{d-1} = 0$,

$$S_{\text{dCFT}} = S_{\text{CFT}} + \lambda \int \mathrm{d}x^0 \, \phi_\alpha(x^0), \tag{1}$$

where $\phi_\alpha(x^0)$ has spin 0 and the scaling dimension $\Delta_\alpha < 1$. For example, this condition is satisfied by the order parameter field [22–27] in the 3D Ising CFT or the $O(N)$ Wilson-Fisher CFT, leading to the definition of a "magnetic line defect" in the respective theories. It is useful to describe such a construction as a perturbation of the "identity" or "trivial" line defect.

By construction, this defect preserves translations in the $x^0$ directions and $SO(d-1)$ rotations in the remaining directions. As the bulk is conformal, the RG flow only affects the line defect. The RG flow is expected to reach a fixed point invariant under the $SO(2,1) \times SO(d-1)$ subgroup of conformal transformations that preserve the shape of the line.[3]

The RG flow of such line defects is controlled by the so-called $g$-theorem, asserting monotonicity of a certain $g$-function of the RG scale [48–53]. The value of $g$ for the fixed point is a non-trivial piece of data. As the trivial line defect has, by definition, $g = 1$, the fixed point of an RG flow starting from the trivial line defect will have $g < 1$. The value of $g$-function is one of the most important quantities that characterizes a line defect. At a fixed point, it can be defined in two ways. The first involves an Euclidean $S^d$ geometry, with the line defect

---

[2]Our discussion does not depend on the original theory having an action, but the assumption makes the presentation a bit more intuitive. One can add extra interactions to any QFT.

[3]This is a general expectation, but not a theorem. In principle, the flow could never reach a fixed point. Examples of that phenomenon can be found even in the Kondo problem. They rely on the existence of bulk operators of dimension exactly 1, which can give marginally irrelevant perturbations supported on a line. If the bulk theory is interacting, such logarithmic behavior of a line defect RG flow should not be possible. Fixed points endowed with scale invariance without conformal invariance should also be non-generic.

wrapping a maximal circle. This geometry is conformally equivalent to the above flat space geometry, but it allows for the definition of a finite partition function and of

$$g = \frac{Z_{\text{dCFT}}}{Z_{\text{CFT}}}. \tag{2}$$

A conformal mapping to $\mathbb{R}^d$ can map the defect to a straight line, depicted in Fig. 1a. The second definition is Lorentzian in nature and relates $g$ to the change of the entanglement entropy of a spatial region surrounding the defect [50, 52]. Both definitions can be employed to prove the $g$-theorem, using either quantum-field-theoretic or quantum-information-theoretic methods. This was first done for $d = 2$ [48–50] and was only recently generalized to higher dimensions [51, 52].

Once we have a defect CFT, we can consider local operators on the line defect, transforming as irreducible representations of the defect conformal group SO(2, 1) [19–21]. A defect primary operator is the lowest operator in these irreps. For defect type $a$,[4] we will denote the primary operator as $\phi_\alpha^{aa}$ and its scaling dimensions as $\Delta_\alpha^{aa}$. The reason for double $a$ in the superscript will be clear shortly. For a nontrivial defect $a$, local operators on the defect have different scaling dimensions from bulk local operators. They also form irreps of the rotational symmetry SO($d-1$).

Given two types of line defects $a$ and $b$, one may join them at a point. There will be a collection of different ways of defining the junction, which can be treated as "defect changing" local operators [54, 55]. As the location of the junction can be moved, one can take $x^0$ derivatives of defect changing local operators. More generally, they can also be organized into irreducible representations of SO(2, 1) × SO($d-1$). We will denote conformal primary defect changing operators as $\phi_\alpha^{ab}$, with scaling dimensions $\Delta_\alpha^{ab}$.

Several special cases of defect changing operators are worth noting. If $b$ is a trivial defect and $a$ is a nontrivial defect, then $\phi_\alpha^{a0}$ is called a defect creation operator (Figure 1b). The physical meaning of defect creation operators can be seen in the following way. If we measure the expectation value of the defect with a finite length $L$ in the bulk CFT, the universal part is given by the two-point correlation function of the defect creation operators, which live at the endpoints of the defect. Thus, $\langle \mathcal{S}^a(L) \rangle = e^{-E L} L^{-2\Delta_0^{a0}}$ where $e^{-E L}$ is a non-universal contribution from the "ground state energy" $E$ of the line defect, which should be subtracted by a local counterterm to obtain a scale-invariant system.

Another important type of defect changing operator corresponds to the domain walls of a global symmetry on the defect line. For simplicity, we consider a CFT with $\mathbb{Z}_2$ global symmetry and a defect created by switching on a $\mathbb{Z}_2$-odd relevant perturbation $\phi_\alpha$. Having $\lambda > 0$ and $\lambda < 0$ will result in two different types of defect $+$ and $-$, which are related by the $\mathbb{Z}_2$ symmetry. The defect changing operator $\phi_0^{+-}$ joining these two defects corresponds to the creation of a domain wall on the defect line (Figure 1c). The string operator which corresponds to a domain wall of length $L$ has the expectation value $L^{-2\Delta_0^{+-}}$ determined by the scaling dimension $\Delta_0^{+-}$ of the domain wall operator on the defect. The scaling dimension is important because it is related to the stability of spontaneous symmetry breaking (SSB) on the defect line. The SSB defect is stable if $\Delta_0^{+-} > 1$. A detailed discussion will be given in Section 6.

Given the importance of dCFTs, it is natural to ask how universal data such as $g$ and scaling dimensions can be computed.

Despite the high degree of symmetry of conformal defects, the full zoo of conformal defects in any given bulk CFT is potentially vast and very poorly understood. Essentially, the only conformal defects for which classification is available are boundary conditions in 2D min-

---

[4]We require $a$ to label a simple defect that cannot be decomposed into a direct sum of two conformal defects. An example of a defect that is not simple is the spontaneous symmetry-breaking defect described below.

imal models [15].[5] Even conformal interfaces in 2d can only be rarely classified [57, 58]. Conversely, for $p > 1$, it is easy to find supersymmetric examples where a classification of conformal defects would be as difficult as a classification of supersymmetric CFTs in dimension $p$. The absence of 1d CFTs gives hope that line defects of any given CFT may yet admit a classification.

The technique of the conformal bootstrap has allowed a highly accurate non-perturbative characterization of many CFTs [59, 60]. The underlying idea is to constrain the conformal data with a set of consistency conditions known as crossing equations. One successful example is the well-known 3D Ising CFT which describes the $\mathbb{Z}_2$ symmetry-breaking transition [61, 62]. The bootstrap program can be extended to defect CFTs, but it requires a more complex set of consistency conditions that combine both bulk and defect local operators [30, 37].

The route that we will take to tackle the problem of dCFTs is based on the numerical simulation of a UV-regularized many-body system (usually a lattice model) at the critical point. Broadly speaking, two important challenges have to be addressed for this approach to work.

The first problem is to establish a connection between the numerically accessible data and the (defect) conformal data. Ground state correlation functions are easy to compute, but it is usually hard to identify CFT operators in the regularized theory, thus limiting its access to the full conformal data. Alternatively, one may make use of the wavefunctions of low-energy eigenstates, which are in one-to-one correspondence of CFT operators due to the state-operator correspondence [63–66]. In $1 + 1$ dimensions, it has been recently demonstrated that overlaps between low-energy wavefunctions of different boundary conditions encode all conformal data [67–69]. The overlap is also easy to compute numerically, making it a simple and systematic approach to extracting conformal data.

The second problem is to deal with finite-size corrections. Usually, a large-scale simulation of a lattice model based on Monte Carlo or tensor networks is required. In more than $1 + 1$ dimensions, obtaining the low-energy eigenstates is extremely nontrivial, limiting the success of the traditional methods. A recent breakthrough in this direction is fuzzy sphere regularization [70], a way to realize the 3D CFTs on the cylinder geometry $S^2 \times \mathbb{R}$ with a small finite-size correction that can already be reached by exact diagonalization. The fuzzy sphere regularization allows direct access to a wealth of information of CFTs [71, 72], and can be applied to various 3D CFTs [73, 74]. More importantly, it has also been extended to obtain the spectrum of defect operators for the 3D Ising model with the magnetic line defect [27].

In this work, we generalize the technique of wavefunction overlap to dCFTs in arbitrary dimensions and apply it to the fuzzy sphere regularization of the 3D Ising CFT. We compute the defect conformal data of the magnetic line defect. Through the overlap of low-energy eigenstates of different defect sectors, we obtain universal information such as the $g$-function, scaling dimensions of defect changing operators, and the OPE coefficients involving these operators. Many of these results are reported for the first time. Furthermore, each answer may be extracted with many different overlaps that are related by conformal invariance. In this way, we can cross-check the consistency of our numerical results.

This work is organized as follows. In Section 2 we summarise our method and results. In Section 3 we review the definition of defect conformal data. In Section 4 we review the state-operator correspondence and derive the relation between wavefunction overlaps and defect conformal data. In Section 5 we apply the formalism to the 3D Ising model with the fuzzy sphere regularization and compute the defect conformal data. In Section 6, we discuss the stability of the Ising SSB defects. Finally, we conclude in Section 7 with discussions and future directions.

---

[5]Boundary conditions in other 2d Rational CFTs can be classified if they preserve a full copy of the chiral algebra of the theory, as solutions of an intricate set of constraints using the underlying algebraic structure (i.e., modular tensor category [56]) of the CFT [16]

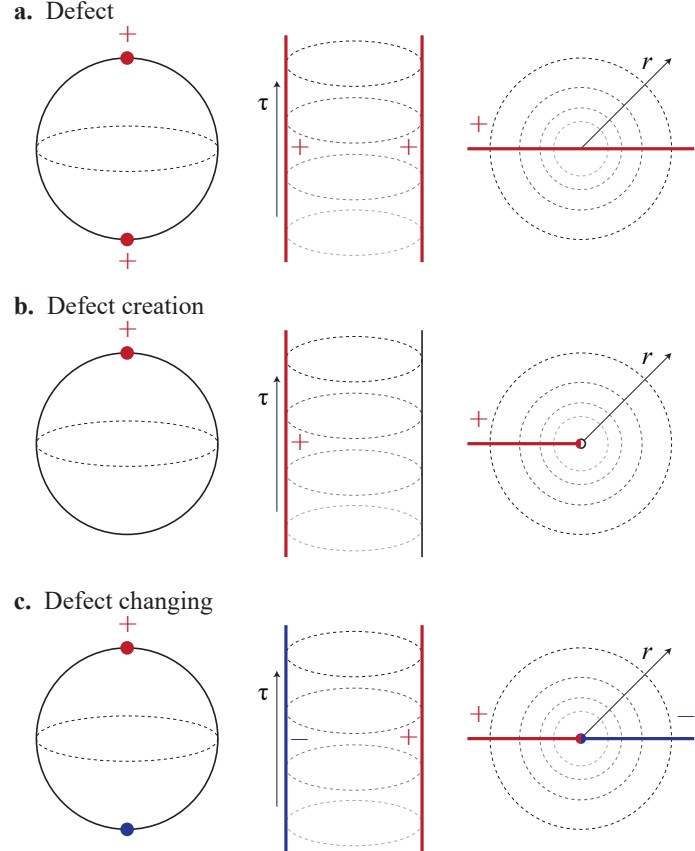

Figure 2: The real space configuration (left column), path integral configuration on cylinder $S^{d-1} \times \mathbb{R}$ (middle column) and Weyl-transformed configuration on flat spacetime $\mathbb{R}^d$ (right column) of (a) an infinitely long line defect (b) defect creation operator and (c) defect changing operator.

*Note: While preparing this manuscript, we became aware of upcoming work [75] studying the magnetic line defect in the 3D Ising CFT using conformal bootstrap.*

## 2 Summary of results

In this paper, we work with an $\mathbb{R} \times S^{d-1}$ cylinder geometry, possibly with line defects located at the poles of the sphere. We start from the CFT Hamiltonian $H_{\text{CFT}}$ on the sphere and add relevant perturbations to the north and south poles

$$H^{ab} = H_{\text{CFT}} + \lambda \mathcal{O}_b(\text{N}) + \lambda' \mathcal{O}_a(\text{S}), \tag{3}$$

where the perturbation $\mathcal{O}_a$ gives rise to defect type $a$ and $\mathcal{O}_b$ gives rise to defect type $b$. A Weyl transformation $(\hat{n}, \tau) \in S^{d-1} \times \mathbb{R} \mapsto e^{\tau/R} \hat{n} \in \mathbb{R}^d$ maps the cylinder to flat spacetime and the support of the line defects to two opposite half-infinite rays (Figure 2). If $a = b$ then we obtain the defect CFT of defect type $a$ with a local operator insertion at the origin, while $a \neq b$ corresponds to configurations with defect changing operators. This is the state-operator map: defect operators $\phi_\alpha^{aa}$ correspond to eigenstates $|\phi_\alpha^{aa}\rangle$ of the defect Hamiltonian $H^{aa}$, and defect changing operators $\phi_\alpha^{ab}$ correspond to eigenstates $|\phi_\alpha^{ab}\rangle$ of the Hamiltonian $H^{ab}$.

Table 1: A summary of our numerical results. (a) The $g$-function of the magnetic line defect in the 3D Ising CFT, the scaling dimension of the lowest defect creation operator $\Delta_0^{+0}$ and the lowest defect changing operator $\Delta_0^{+-}$, and the OPE coefficients $C_{000}^{+0-} : \phi_0^{+0} \times \phi_0^{0-} \to \phi_0^{+-}$. (b) The creation-creation-bulk OPE coefficients $C_{00\alpha}^{0+0} : \phi_0^{+0} \times \phi_0^{+0} \to \phi_\alpha^{00}$, creation-creation-defect OPE coefficients $C_{00\alpha}^{+0+} : \phi_0^{+0} \times \phi_0^{+0} \to \phi_\alpha^{++}$, and changing-changing-defect OPE coefficients $C_{00\alpha}^{+-+} : \phi_0^{+-} \times \phi_0^{-+} \to \phi_\alpha^{++}$. The error bars are estimated from finite size extrapolation, which we will discuss in Section 5.3. The error bars of $\Delta_0^{+0}, \Delta_0^{+-}$ and $C_{000}^{+0-}$ also come from averaging values from different methods. We also need to note that the error bars given are not strict but only an estimation.

(a)

| Quantity | Result |
|---|---|
| $g$ | 0.602(2) |
| $\Delta_0^{+0}$ | 0.108(5) |
| $\Delta_0^{+-}$ | 0.84(5) |
| $C_{000}^{+0-}$ | 0.77(5) |

(b)

| $\phi_\alpha^{00}$ | $C_{00\alpha}^{0+0}$ | $\phi_\alpha^{++}$ | $C_{00\alpha}^{+0+}$ | $C_{00\alpha}^{+-+}$ |
|---|---|---|---|---|
| $\sigma$ | 0.869(19) | $\phi_1^{++}$ | 0.22(3) | 1.25(12) |
| $\epsilon$ | 0.3334(9) | $\phi_2^{++}$ | 0.053(19) | 1.01(26) |
| $T^{\mu\nu}$ | 0.044(28) | $\phi_3^{++}$ | / | / |
| $\epsilon'$ | 0.003(7) | $\phi_4^{++}$ | 0.007(3) | 0.009(5) |

As all the eigenstates exist within the same Hilbert space,[6] the overlap between these states can be expressed to the leading order as a four-point correlator

$$
A_{\alpha\beta}^{abcd} = \langle \phi_\beta^{cd} | \phi_\alpha^{ab} \rangle
$$
$$
= M_0^{ca} M_0^{bd} \left( \frac{1}{R} \right)^{\Delta_0^{ca} + \Delta_0^{bd}} e^{-(\delta_{ab} \log g_a + \delta_{cd} \log g_c)/2} \langle \phi_0^{ca}(-1) \phi_\alpha^{ab}(0) \phi_0^{bd}(1) \phi_\beta^{dc}(\infty) \rangle_{\text{pl}}, \quad (4)
$$

where $M_0^{ca}, M_0^{bd}$ are non-universal constants, $g_a$ and $g_c$ are the $g$-functions of the different types of defects. Defect conformal data can be obtained by taking ratios of different overlaps, where the non-universal pieces cancel out.

Specifically, we consider the magnetic line defect in the 3D Ising CFT, where $a, b$ are taken as $+, -$ denoting adding a positive or negative $\sigma$ operator, or 0 denoting not adding a defect at the points. Through the equation, we can extract a wealth of conformal data, including

1. The $g$-function of the magnetic line defect in the 3D Ising CFT ;

2. The scaling dimensions of defect creation operators $\Delta_\alpha^{+0}$ and the defect changing operators $\Delta_\alpha^{+-}$. In particular, we find $\Delta_0^{+-} = 0.84(4) < 1$, so the spontaneous Ising symmetry-breaking defect is unstable.

3. The OPE coefficients $C_{\alpha\beta\gamma}^{abc} : \phi_\alpha^{ab} \times \phi_\beta^{bc} \to \phi_\gamma^{ac}$, where all the operators are collinear. Specifically, we have computed creation-creation-bulk $C_{00\alpha}^{0+0} : \phi_0^{+0} \times \phi_0^{+0} \to \phi_\alpha^{00}$, creation-creation-defect $C_{00\alpha}^{+0+} : \phi_0^{+0} \times \phi_0^{+0} \to \phi_\alpha^{++}$, change-change-defect $C_{00\alpha}^{+-+} : \phi_0^{+-} \times \phi_0^{+-} \to \phi_\alpha^{++}$, and creation-creation-change $C_{000}^{+0-} : \phi_0^{+0} \times \phi_0^{-0} \to \phi_0^{+-}$.

4. We reproduce the scaling dimensions of primary operators of the bulk Ising CFT and the dCFT using the overlap method. Our results are all within 1.5% discrepancy with previous results based on bulk conformal bootstrap [62] or energy levels on fuzzy sphere [27].

Our main new results are listed in Table 1.

---

[6]In field theory this statement is only true in a specific regularization scheme before the UV regulator is sent to 0. The overlaps go to zero as the regulator is removed, but the ratios we consider remain finite and scheme-independent. Here we work with a specific UV completion given by the fuzzy sphere regularization.

# 3 Defect conformal kinematics

As the bulk CFT, a defect CFT is also governed by a set of universal conformal data. Apart from the $g$-function, the rest of the conformal data in a defect CFT could be determined by the correlation functions on the defect, which are constrained by conformal symmetry $SO(2, 1)$ in 1D. The defect operators and the defect changing operators then transform according to the irreducible representation. Both defect operators and, perhaps more surprisingly, defect changing operators can be organized into conformal towers, consisting of the primary operators $\phi_\alpha^{ab}$ with scaling dimensions $\Delta_\alpha^{ab}$ and their descendants $\partial_0^n \phi_\alpha^{ab}$ with scaling dimension $\Delta_\alpha^{ab} + n$ generated by acting with the derivative along the defect. An important special property of the "$aa$" defect operator spectrum is that it contains at least one copy of an identity operator $\phi_0^{aa}$, with $\Delta_0^{aa} = 0$ and $\partial \phi_0^{aa} = 0$.

The two-point correlation function of primaries on defects is determined by the scaling dimension:

$$\langle \phi_\alpha^{ab}(x_1^0) \phi_\beta^{ba}(x_2^0) \rangle = |x_{12}^0|^{-2\Delta_\alpha^{ab}} \delta_{\alpha\beta} , \tag{5}$$

where $x_{12}^0 = x_1^0 - x_2^0$. The Hermitian conjugation of a primary operator is given by the inversion,

$$(\phi_\alpha^{ab}(x^0))^\dagger = |x^0|^{-2\Delta_\alpha^{ab}} \phi_\alpha^{ba}(1/x^0). \tag{6}$$

Thus, one can define

$$\phi_\alpha^{ba}(\infty) := (\phi_\alpha^{ab}(0))^\dagger = \lim_{x^0 \to \infty} |x^0|^{2\Delta_\alpha^{ab}} \phi_\alpha^{ba}(x^0), \tag{7}$$

such that $\langle \phi_\alpha^{ab}(0) \phi_\alpha^{ba}(\infty) \rangle = 1$. The operator product expansion can also be applied to defect operators and defect changing operators. For example, consider a line with three consecutive segments of defects $a, b, c$, with two defect changing operators $\phi_\alpha^{ab}$ and $\phi_\beta^{bc}$. If the length of the defect $b$ is small, the two defect changing operators can be fused into a linear combination of defect changing operators from $a$ to $c$,

$$\phi_\alpha^{ab}(x_1^0) \phi_\beta^{bc}(x_2^0) = \sum_\gamma C_{\alpha\beta\gamma}^{abc} |x_{12}^0|^{\Delta_\gamma^{ac} - \Delta_\alpha^{ab} - \Delta_\beta^{bc}} \phi_\gamma^{ac}(x_2^0, 0) + \text{des.} , \tag{8}$$

where des. means descendant operators whose coefficient is completely fixed by the conformal symmetry. In particular, if $a$ and $c$ are the same type of defect, then the fusion gives rise to a defect operator. If $a$ and $c$ are trivial defects the fusion gives bulk operator without any defect line, though still organized in line defect conformal towers. The OPE coefficient determines the three-point correlation function of defect changing operators.

$$\langle \phi_\alpha^{ab}(x_1^0) \phi_\beta^{bc}(x_2^0) \phi_\gamma^{ca}(x_3^0) \rangle = C_{\alpha\beta\gamma}^{abc} |x_{12}^0|^{\Delta_\gamma^{ac} - \Delta_\alpha^{ab} - \Delta_\beta^{bc}} |x_{23}^0|^{\Delta_\alpha^{ab} - \Delta_\beta^{bc} - \Delta_\gamma^{ac}} |x_{13}^0|^{\Delta_\beta^{bc} - \Delta_\alpha^{ab} - \Delta_\gamma^{ac}} . \tag{9}$$

Two special cases we will be using are

$$\langle \phi_\alpha^{ab}(0) \phi_\beta^{bc}(1) \phi_\gamma^{ca}(\infty) \rangle = C_{\alpha\beta\gamma}^{abc} , \tag{10}$$

$$\langle \phi_\alpha^{ab}(-1) \phi_\beta^{bc}(0) \phi_\gamma^{ca}(1) \rangle = C_{\alpha\beta\gamma}^{abc} 2^{\Delta_\beta^{bc} - \Delta_\alpha^{ab} - \Delta_\gamma^{ac}} . \tag{11}$$

If one of the three operators is the identity operator $I = \phi_0^{aa}$, the three-point correlation function reduces to a two-point correlation function. In this work, we use the convention such that those OPEs are normalized, i.e., $C_{\alpha\beta 0}^{aba} = \delta_{\alpha\beta}$. We especially note that for the three-point function of two defect creation operators with a bulk CFT operator, we only consider the case where the three points are on the same line. Otherwise, the form of the correlation function could not be completely fixed by conformal symmetry.

To obtain a complete characterization of the defect CFT, one needs additional data known as bulk-to-defect OPEs, which determine correlation functions involving both bulk operators and defect operators. These OPEs have been computed before for the three-dimensional Ising model in Ref. [27]. The bulk-to-defect OPEs, together with the defect data including $g$-function $g_a$, scaling dimensions $\Delta_\alpha^{ab}$ and the defect OPEs $C_{\alpha\beta\gamma}^{abc}$ satisfy a number of constraints. In $1+1$ dimensional rational CFTs with rational defects, the constraints can be solved with the modular tensor category data. In higher dimensions, no such algebraic structure is available. In this work, we show how they can be computed using the wavefunction overlap method in arbitrary dimensions. We will apply the method to the 3D Ising CFT with a magnetic line defect.

# 4 Wavefunction overlap in defect CFT

## 4.1 State-operator correspondence of defect CFT

We first review some basic facts about CFTs with no defect. We start with a $d$-dimensional CFT on the cylinder $\mathbb{R} \times S^{d-1}$, where the time coordinate $\tau \in (-\infty, \infty)$ and the spatial coordinate $\Omega \in S^{d-1}$. The metric is

$$ds_{\text{cyl}}^2 = d\tau^2 + \tilde{R}^2 d\Omega^2, \tag{12}$$

where $\tilde{R}$ is the radius of the sphere and $d\Omega^2$ is the surface element on $S^{d-1}$. For this work, we will consider $d = 3$, where $d\Omega^2 = d\theta^2 + \sin^2\theta d\phi^2$ with the spherical coordinates $(\theta, \phi)$ on $S^2$. On the other hand, the CFT may be defined on a Euclidean flat spacetime $\mathbb{R}^d$, with metric

$$ds_{\text{pl}}^2 = \sum_{\alpha=0}^{d-1} (dx^\alpha)^2. \tag{13}$$

In the spherical coordinates, the metric can be alternatively written as

$$ds_{\text{pl}}^2 = dr^2 + r^2 d\Omega^2, \tag{14}$$

where

$$r := \sqrt{\sum_{\alpha=0}^{d-1} (x^\alpha)^2}. \tag{15}$$

The two metrics are related by a conformal transformation

$$\tau = \log \frac{r}{\tilde{R}}, \tag{16}$$

$$ds_{\text{pl}}^2 = f(r)^2 ds_{\text{cyl}}^2, \tag{17}$$

$$f(r) = \frac{r}{\tilde{R}}, \tag{18}$$

where $f(r)$ is the conformal factor.

The state-operator correspondence is the statement that an eigenstate $|\phi_\alpha\rangle$ of the CFT Hamiltonian $H_{\text{CFT}}$ on $S^{d-1}$ is in one-to-one correspondence with the CFT operator $\phi_\alpha(\tau, \Omega)$ by

$$|\phi_\alpha\rangle = \phi_\alpha(\tau = -\infty)|\mathbb{I}^{00}\rangle, \tag{19}$$

where $|\mathbb{I}^{00}\rangle \equiv |\mathbb{I}\rangle$ is the ground state without any defect, and

$$\phi_\alpha(\tau = -\infty) := \tilde{R}^{\Delta_\alpha} \lim_{\tau \to -\infty} e^{-\Delta_\alpha \tau/\tilde{R}} \phi_\alpha(\tau, \Omega). \tag{20}$$

For an intuitive derivation of formulae involving wave function overlaps, we will represent the wave function of a state as a path integral with an operator insertion at $\tau = -\infty$:

$$\langle \Phi(\Omega)|\phi_\alpha\rangle = \frac{1}{\sqrt{Z_{\text{CFT}}}} \int_{\Phi(\tau=0,\Omega)=\Phi(\Omega)} \mathcal{D}\Phi \, \phi_\alpha(\tau=-\infty)e^{-S_{\text{CFT}}[\Phi]}, \tag{21}$$

where $\{|\Phi(\Omega)\rangle\}$ is a coherent state basis state of the CFT, and

$$Z_{\text{CFT}} = \int \mathcal{D}\Phi \, e^{-S_{\text{CFT}}[\Phi]}, \tag{22}$$

is the partition function of the CFT on $\mathbb{R} \times S^{d-1}$. Analogously, the bra state is defined with an insertion at $\tau = +\infty$,

$$\langle \phi_\alpha| = \langle \mathbb{I}^{00}|\phi_\alpha(+\infty), \tag{23}$$

where

$$\phi_\alpha(+\infty) := \tilde{R}^{\Delta_\alpha} \lim_{\tau\to+\infty} e^{\Delta_\alpha \tau/\tilde{R}} \phi_\alpha(\tau,\Omega). \tag{24}$$

Furthermore, the energy $E_\alpha$ of the eigenstate corresponds to the scaling dimension by

$$E_\alpha = \frac{\Delta_\alpha}{\tilde{R}}. \tag{25}$$

Now let us consider a path integral of the defect CFT. Under the conformal transformation, the defect line $x^1 = x^2 = \cdots = x^{d-1} = 0$ is mapped to two lines on $\mathbb{R} \times S^{d-1}$, which are parallel to the time direction and cross north pole N or south pole S. The line crossing the north pole corresponds to $x^0 > 0$ and the line crossing the south pole corresponds to $x^0 < 0$. The Hamiltonian of $S^{d-1}$ has two insertions at the north pole and the south pole,

$$H^{aa} = H_{\text{CFT}} + \lambda \mathcal{O}_a(\text{N}) + \lambda \mathcal{O}_a(\text{S}). \tag{26}$$

An eigenstate $|\phi_\alpha^{aa}\rangle$ of $H_{\text{dCFT}}$ corresponds to a defect operator $\phi_\alpha^{aa}(\tau)$ by

$$|\phi_\alpha^{aa}\rangle = \phi_\alpha^{aa}(\tau=-\infty)|\mathbb{I}^{aa}\rangle. \tag{27}$$

In terms of path integral, the wave-functional is

$$\langle \Phi(\Omega)|\phi_\alpha^{aa}\rangle = \frac{1}{\sqrt{Z^{aa}}} \int_{\substack{\Phi(\tau=0,\Omega)=\Phi(\Omega) \\ \tau<0}} \mathcal{D}\Phi \, \phi_\alpha^{aa}(\tau=-\infty)e^{-S_{\text{dCFT}}[\Phi]}, \tag{28}$$

where

$$\phi_\alpha^{aa}(\tau=-\infty) := \tilde{R}^{\Delta_\alpha^{aa}} \lim_{\tau\to-\infty} e^{-\Delta_\alpha^{aa}\tau/\tilde{R}} \phi_\alpha^{aa}(\tau). \tag{29}$$

Recall that the $aa$ sector contains a canonically normalized identity operator[7] $\phi_0^{aa}$ whose two-point function is related to the $g$-function of the defect as in Eq. (2). Accordingly, we choose a normalization constant $Z^{aa} = g_a Z_{\text{CFT}}$. This choice will be convenient for extracting the $g$-function from the overlap. The bra state is again defined by inserting the operator at $\tau = +\infty$.

Finally, we consider the defect changing operators $\phi_\alpha^{ab}$, which include defect creation operators as a special case ($b=0$). These operators correspond to eigenstates $|\phi_\alpha^{ab}\rangle$ of the Hamiltonian

$$H^{ab} = H_{\text{CFT}} + \lambda \mathcal{O}_b(\text{N}) + \lambda' \mathcal{O}_a(\text{S}), \tag{30}$$

---

[7]Normalized so that any operator fusing with $\phi_0^{aa}$ gives exactly the operator itself.

where the perturbation $\mathcal{O}_a$ gives rise to defect type $a$ and $\mathcal{O}_b$ gives rise to defect type $b$. The path integral representation is given by

$$\langle \Phi(\Omega)|\phi_\alpha^{ab}\rangle = \frac{1}{\sqrt{Z^{ab}}} \int_{\substack{\Phi(\tau=0,\Omega)=\Phi(\Omega) \\ \tau<0}} \mathcal{D}\Phi\, \phi_\alpha^{ab}(\tau=-\infty)e^{-S^{ab}[\Phi]}, \tag{31}$$

where

$$S^{ab} = S_{\text{CFT}} + \lambda \int d\tau\, \mathcal{O}_b(\tau, \text{N}) + \lambda' \int d\tau\, \mathcal{O}_a(\tau, \text{S}), \tag{32}$$

is the CFT action with two defect lines inserted. The path integral for the bra state is

$$\langle \phi_\alpha^{ab}|\Phi(\Omega)\rangle = \frac{1}{\sqrt{Z^{ab}}} \int_{\substack{\Phi(\tau=0,\Omega)=\Phi(\Omega) \\ \tau>0}} \mathcal{D}\Phi\, \phi_\alpha^{ba}(\tau=+\infty)e^{-S^{ab}[\Phi]}, \tag{33}$$

where the order of $a$ and $b$ changes because $(\phi_\alpha^{ab}(\tau=-\infty))^\dagger = \phi_\alpha^{ba}(\tau=+\infty)$. Lacking a canonically-normalized identity operator, we choose $Z^{ab} = Z_{\text{CFT}}$ for $a \neq b$, which amounts to the normalization of defect changing operator $\phi_\alpha^{ab}$ such that $\langle \phi_\alpha^{ab}(\tau=-\infty)\phi_\alpha^{ba}(\tau=+\infty)\rangle = 1$. For a unified notation, we may write

$$Z^{ab} = Z_{\text{CFT}}e^{-\delta_{ab}\log g_a}. \tag{34}$$

The factor $Z_{\text{CFT}}$ will cancel in all physical quantities we compute, so we set it to 1 in the formulae below.

## 4.2 Path integral for wavefunction overlap

We consider the overlap

$$A_{\alpha\beta}^{abcd} = \langle \phi_\beta^{cd}|\phi_\alpha^{ab}\rangle. \tag{35}$$

Taking the overlap amounts to gluing the path integral for the bra and the ket on the $\tau=0$ surface. Equivalently, this can be represented by insertion of the resolution of identity

$$A_{\alpha\beta}^{abcd} = \int \mathcal{D}\Phi(\tau=0,\Omega)\, \langle \phi_\beta^{cd}|\Phi(\tau=0,\Omega)\rangle\langle \Phi(\tau=0,\Omega)|\phi_\alpha^{ab}\rangle, \tag{36}$$

where the integration measure $\mathcal{D}\Phi$ only includes the field configurations on the $\tau=0$ surface. Thus, the path integral for the overlap is the CFT path integral on $\mathbb{R} \times S^{d-1}$ with four defect

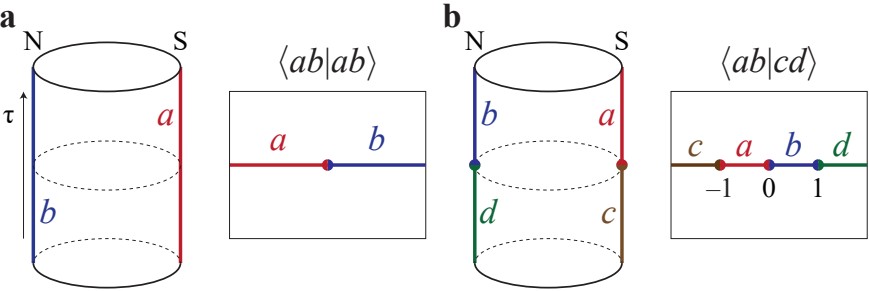

Figure 3: An illustration of the path integral configuration on $\mathbb{R} \times S^{d-1}$ (left) and the corresponding configurations in flat spacetime $\mathbb{R}^d$ after Weyl transformation (right) for different overlaps.

lines inserted, see Figure 3b. More explicitly, we can substitute Eqs. (33) and (31) into Eq. (36) to obtain

$$A^{abcd}_{\alpha\beta} = \frac{1}{\sqrt{Z^{ab}Z^{cd}}} \langle \mathcal{O}^{ca}(\tau=0,\mathrm{S})\phi^{ab}_{\alpha}(\tau=-\infty)\mathcal{O}^{bd}(\tau=0,\mathrm{N})\phi^{dc}_{\beta}(\infty)\rangle_{\mathrm{cyl}}. \quad (37)$$

The operator $\mathcal{O}^{ca}$ and $\mathcal{O}^{bd}$ are defect changing operators inserted on the north pole and south pole at the $\tau=0$ time slice. They must be inserted if the defect types $a$ and $c$ (or $b$ and $d$) are not the same. It is natural to expand these operators in terms of the scaling operators in the defect changing sector,

$$\begin{aligned} \mathcal{O}^{ca} &= \sum_{\gamma} M^{ca}_{\gamma} \epsilon^{\Delta^{ca}_{\gamma}} \phi^{ca}_{\gamma}, \\ \mathcal{O}^{bd} &= \sum_{\delta} M^{bd}_{\gamma} \epsilon^{\Delta^{bd}_{\delta}} \phi^{bd}_{\delta}, \end{aligned} \quad (38)$$

where $\epsilon$ is a short-distance regulator. The power of $\epsilon$ ensures that each term in the sum has the same eigenvalue under rescaling. $M^{ca}_{\gamma}, M^{bd}_{\delta}$ are non-universal constants, as the detail of gluing the two defects relies on the UV realization. In the continuum limit, the leading term with the smallest scaling dimension will dominate. If $a$ and $c$ are the same type of defect, then nothing happens at the intersection point. This corresponds to inserting an identity operator $I = \phi^{aa}_0$ and setting the normalization $M^{aa}_0 = 1$. Otherwise, the insertion $\mathcal{O}^{ca}$ has a nonzero scaling dimension which corresponds to the most relevant defect changing operator $\phi^{ca}_0$ allowed by symmetry. In particular, the operator insertion is constrained by the $\mathrm{SO}(d-1)$ rotation symmetry. In $d=3$, the representation of the operators $\phi^{ab}_{\alpha}$ under $\mathrm{SO}(d-1)$ rotational symmetry is given by the angular momentum along $z$-axis $(L^z)^{ab}_{\alpha}$, and the constraint becomes the angular momentum conservation $(L^z)^{ab}_{\alpha} + (L^z)^{ca}_{\gamma} + (L^z)^{bd}_{\delta} = (L^z)^{cd}_{\beta}$.

Now we use the conformal transformation to obtain a correlation function in $\mathbb{R}^d$. The transformation for primary operators from $\mathbb{R} \times S^{d-1}$ to $\mathbb{R}^d$ is $\phi^{ab}_{\alpha}(\tau,\Omega) \to f(r,\Omega)^{\Delta^{ab}_{\alpha}} \phi^{ab}_{\alpha}(r,\Omega)$. Thus, combining Eqs. (37) and (34) we obtain

$$A^{abcd}_{\alpha\beta} = M^{ca}_0 M^{bd}_0 \left(\frac{1}{R}\right)^{\Delta^{ca}_0 + \Delta^{bd}_0} e^{-(\delta_{ab}\log g_a + \delta_{cd}\log g_c)/2} \langle \phi^{ca}_0(-1)\phi^{ab}_{\alpha}(0)\phi^{bd}_0(1)\phi^{dc}_{\beta}(\infty)\rangle_{\mathrm{pl}}. \quad (39)$$

The dependence on the radius $R = \tilde{R}/\epsilon$ appears because of the conformal factor. For an actual finite-size calculation, $R$ is the ratio between the UV scale and the IR scale. It is important to note that $\alpha$ labels a primary operator. Overlaps involving descendant states give nontrivial cross-checks for this relation. We will consider them in the appendix.

Eq. (39) explicitly relates the wavefunction overlap with universal quantities such as the $g$-function and the four-point correlation function. Yet, it still contains non-universal coefficients $M^{ca}_0$, $M^{bd}_0$ in front of them. In the next subsection, we consider the ratio of different overlaps to extract the conformal data, where the non-universal pieces exactly cancel out.

## 4.3 Defect conformal data from wavefunction overlap

As reviewed in Section 3, defect conformal data includes the $g$-function $g_a$, scaling dimension of defect operators $\Delta^{aa}_{\alpha}$ and defect changing operators $\Delta^{ab}_{\alpha}$, and the OPE coefficients $C^{abc}_{\alpha\beta\gamma}$. We will show that how the conformal data above can be obtained by taking ratios of overlaps. We note that for the OPE coefficients $C^{abc}_{\alpha\beta\gamma}$, we require one of the three operators to be the lowest primary operator. As it turns out, each conformal data may be computed using many different overlaps, which serve as cross-checks of each other.

First, we extract the $g$-function. For this purpose, it is useful to simplify the four-point correlation functions by demanding two operators to be $I = \phi^{aa}_0$. One way to achieve the goal

is to take the ratio of the following two overlaps,

$$g_a = \left( \frac{A_{00}^{a000}}{A_{00}^{a0aa}} \right)^2 . \tag{40}$$

The non-universal coefficients cancel because $M_0^{aa} = 1$ and the correlation function cancel because $\langle \phi^{a0}(0)\phi^{0a}(1)\rangle = \langle \phi^{0a}(-1)\phi^{a0}(0)\rangle = 1$.

Next, we extract the scaling dimension of the leading defect changing operators $\Delta_0^{ac}$, where $a \neq c$. One way is to look at the scaling of the overlap with the size $R$, $A_{00}^{aaac} \sim R^{-\Delta_0^{ac}}$, i.e.,

$$\Delta_0^{ac} = -\frac{d \log A_{00}^{aaac}}{d \log R} . \tag{41}$$

If we only have access to a discrete set of $R$'s, which is the case for the fuzzy sphere calculation later, $\Delta_0^{ac}$ can be obtained by a linear regression. Alternatively, one may set $a = b$ and $c = d$, and

$$\Delta_0^{ac} = -\frac{1}{2} \frac{d \log A_{00}^{aacc}}{d \log R} . \tag{42}$$

The prefactor $1/2$ is due to the two insertions of defect changing operators at N and S on the $\tau = 0$ slice.

At this point, there is a nontrivial cross-check from

$$\frac{A_{00}^{aabb}}{(A_{00}^{aaab})^2} = 2^{-2\Delta_0^{ab}} \sqrt{\frac{g_a}{g_b}} , \tag{43}$$

which is independent of $R$. The prefactor comes from $\langle \phi_0^{ab}(-1)\phi_0^{ba}(1)\rangle = 2^{-2\Delta_0^{ab}}$.

For the higher leading defect changing operator $\Delta_\alpha^{ac}$, one may consider the overlap $A_{0\alpha}^{aaac}$. Here the leading correlation function in Eq. (37) vanishes because the two-point correlation function $\langle \phi_0^{ac}(\tau = 0, N)\phi_\alpha^{ca}(\infty)\rangle = 0$ for $\alpha \neq 0$. Instead, the leading contribution comes from the $\phi_\alpha^{ac}$ term in the expansion Eq. (38). Thus the overlap $A_{0\alpha}^{aaac}$ scales as $R^{-\Delta_\alpha^{ac}}$ due to the conformal factor. In this way, we can read off the scaling dimensions of higher primary defect changing operators.

The OPE coefficient $C_{\alpha\beta\gamma}^{abc}$ is given by three-point correlation functions, which can be obtained from Eq. (39) by setting one operator to the identity. This scheme can only give a subset of the OPE coefficients, namely one of the three operators should be the lowest defect changing operator, e.g., $\phi_0^{bc}$ with $b \neq c$. This is because the correlation function in Eq. (39) only contains the lowest defect changing operators at each of the intersections. The ratio to compute is

$$\frac{A_{\alpha\gamma}^{abac}}{A_{00}^{bbbc}} = C_{\alpha0\gamma}^{abc} e^{(\log g_b - (\delta_{ab} + \delta_{ac})\log g_a)/2} . \tag{44}$$

The OPE coefficient comes from Eq. (10). Note that the OPE coefficient is cyclic in its arguments, $C_{\alpha\beta\gamma}^{abc} = C_{\beta\gamma\alpha}^{bca}$. Thus, a single OPE coefficient may be computed from many different overlaps, which serve as cross-checks of each other.

Finally, we consider the scaling dimensions $\Delta_\gamma^{aa}$ of defect operators $\phi_\gamma^{aa}$, which also include the bulk operators for $a = 0$ (trivial defect). The overlap ratio is

$$\frac{A_{\gamma0}^{aabb}}{A_{00}^{aabb}} = C_{0\gamma0}^{baa} 2^{\Delta_\gamma^{aa}} , \tag{45}$$

where $a \neq b$ and we have used Eq. (11). Note that the OPE coefficient $C_{0\gamma0}^{baa} = C_{00\gamma}^{aba}$ can be computed from Eq. (44) by setting $c = a$ and $\alpha = 0$.

### 4.4 Finite-size corrections

Before concluding this section, we comment on finite-size corrections to Eqs. (40)-(45) above. As noted in Eq. (38), the intersection of two different types of defect contains the contributions from all symmetry-allowed defect changing operators, including primary operators and descendant operators. Specifically, from Eqs. (37) and (38), the dependence of the overlap $A_{\alpha\beta}^{abcd}$ on the system size can be expressed as

$$A_{\alpha\beta}^{abcd}(R) = \text{constant} \times R^{-(\Delta_0^{ca}+\Delta_0^{bd})}\left(1 + \sum_{ij}\text{constant} \times R^{-(\Delta_i^{ca}+\Delta_j^{bd})+(\Delta_0^{ca}+\Delta_0^{bd})}\right). \quad (46)$$

Here $i$ ($j$) is summed over all the symmetry-allowed operators including primaries and descendants that carry the same quantum numbers if $a \neq c$ ($b \neq d$). For identical defect $a = c$ ($b = d$), we always have the identity operator at the insertion. In this case, the sum only involves $i = 0$ ($j = 0$). The constants are independent of system size and rely on the microscopic details of the model.

To fit the finite-size data, we will truncate the sum to a certain scaling dimension, as the contribution from higher defect changing operators is subleading. In the following, we will only keep the contributions from the two lowest defect changing operators.

We consider some examples in the 3D Ising model that will be useful in the following. For overlaps that contain defect-creation operators $\phi_\alpha^{\pm 0}$ on the interface, we will see from the operator spectrum (see Section 5.2) that the subleading contributions come from the first and second descendants of $\phi_0^{+0}$,

$$A_{00}^{+000}(R), A_{00}^{+++0}(R) \sim R^{-\Delta_0^{+0}}\left(1 + \text{constant} \times R^{-1} + \text{constant} \times R^{-2}\right),$$
$$A_{00}^{++00}(R), A_{00}^{+-00}(R) \sim R^{-2\Delta_0^{+0}}\left(1 + \text{constant} \times R^{-1} + \text{constant} \times R^{-2}\right). \quad (47a)$$

For overlaps that contain defect changing operators $\phi_\alpha^{+-}$ on the interface, the subleading contributions come from the second descendants of $\phi_0^{+-}$[8] and the second primary $\phi_1^{+-}$

$$A_{00}^{+++-}(R) \sim R^{-\Delta_0^{+-}}\left(1 + \text{constant} \times R^{-2} + \text{constant} \times R^{-(\Delta_1^{+-}-\Delta_0^{+-})}\right),$$
$$A_{00}^{++--}(R) \sim R^{-2\Delta_0^{+-}}\left(1 + \text{constant} \times R^{-2} + \text{constant} \times R^{-(\Delta_1^{+-}-\Delta_0^{+-})}\right). \quad (47b)$$

We will see that from the state-operator correspondence that $\Delta_1^{+-}-\Delta_0^{+-} \approx 2.4$. Taking the ratio of two overlaps results in finite-size corrections which inherit all of the power law contributions from the numerator and denominator.

We also note that the finite-size extrapolation is not indispensable for most computations. Its purpose is only to further improve the numerical accuracy.

## 5 Defect conformal data of the 3D Ising CFT

### 5.1 Fuzzy sphere

To compute the defect conformal data using the overlap method, we need a UV-finite realization of the CFT on the sphere. In this work, we use the fuzzy sphere regularization [70] of the 3D Ising CFT, which has small finite-size corrections with a small size. The fuzzy sphere realization employs spinful electrons moving on the sphere in the presence of $4\pi s$ units of uniform

---

[8]The first derivative $\partial_0\phi_0^{+-}$ carries a different quantum number than $\phi_0^{+-}$ under the joint action of the $\pi$-rotation around the $x$-axis $\theta, \phi \mapsto \pi - \theta, -\phi$ and the spin-flip $c_\sigma \mapsto c_{-\sigma}$. Thus, it does not contribute.

magnetic flux. Because of the magnetic flux, the single-electron states form quantized Landau levels. There are $N = 2s+1$ degenerate orbitals for each flavor (i.e. spin) in the lowest Landau level (LLL). We half-fill the LLL and set the gap between the LLL and higher Landau levels to be much larger than other energy scales in the system. In this case, we can effectively project the system into the LLL. After the projection, the coordinates of electrons are not commuting anymore. We thus end up with a system defined on a fuzzy (non-commutative) two-sphere, and the linear size of the sphere $R$ is proportional to $\sqrt{N}$. The 3D Ising transition can be realized by two-flavour fermions $\Psi(\vec{r}) = (\psi_\uparrow(\vec{r}), \psi_\downarrow(\vec{r}))^\mathrm{T}$ with interactions that mimic a $(2+1)$D transverse field Ising model on the sphere

$$H_{\text{bulk}} = \int d^2\vec{r}_1 d^2\vec{r}_2\, U(\vec{r}_{12})[n^0(\vec{r}_1)n^0(\vec{r}_2) - n^z(\vec{r}_1)n^z(\vec{r}_2)] - h\int d^2\vec{r}\, n^x(\vec{r}). \tag{48}$$

The density operators are defined as $n^\alpha(\vec{r}) = \Psi^\dagger(\vec{r})\sigma^\alpha\Psi(\vec{r})$, where $\sigma^{x,y,z}$ are the Pauli-matrices and $\sigma^0 = \mathbb{I}_2$. The density-density interaction potential is taken as $U(\vec{r}_{12}) = g_0 R^{-2}\delta(\vec{r}_{12}) + g_1 R^{-4}\nabla^2\delta(\vec{r}_{12})$. The potential $U(\vec{r}_{12})$ and the transverse field $h$ are taken from Ref. [70] to ensure the bulk realizes the Ising CFT. In practice, the second quantized operators can be expressed in terms of the annihilation operator in the orbital space as $\Psi(\vec{r}) = (1/\sqrt{N})\sum_{m=-s}^{s} Y_{sm}^{(s)}(\hat{r})c_m$, where the monopole spherical harmonics

$$Y_{sm}^{(s)}(\hat{r}) = \mathcal{N}_m e^{im\phi}\cos^{s+m}\frac{\theta}{2}\sin^{s-m}\frac{\theta}{2}, \tag{49}$$

is the single particle wavefunction of the LLL [76]. The Hamiltonian can be expressed as quadratic and quartic terms of the creation and annihilation operators $c_m^\dagger, c_m$ as well [77].

To study the magnetic line defect and the defect changing or creation operators, we add point-like magnetic impurities at the north and south pole of the sphere [27]

$$H_{\text{defect}} = -h_\mathrm{N} n^z(\mathrm{N}) - h_\mathrm{S} n^z(\mathrm{S}). \tag{50}$$

Any finite $h_\mathrm{N}, h_\mathrm{S}$ will trigger a defect RG to the infinite coupling fixed point, so we can simply set $h_\mathrm{N}, h_\mathrm{S} \to \pm\infty$. As the density operator at the north and south pole only involves the highest and lowest-spin orbitals $n^z(\mathrm{N}) = c_s^\dagger\sigma^z c_s$, $n^z(\mathrm{S}) = c_{-s}^\dagger\sigma^z c_{-s}$, we take the $h_\mathrm{N} \to +\infty$ limit by setting the $c_{s,\uparrow}$ orbital to be occupied and the $c_{s,\downarrow}$ orbital to be empty. The $h_\mathrm{N} \to -\infty$ and the $h_\mathrm{S} \to \pm\infty$ limits can be taken similarly. We denote the limit $h_{\mathrm{N,S}} \to \pm\infty$ or $h_{\mathrm{N,S}} = 0$ by a superscript $+, -$ or $0$ on the Hamiltonian, e.g., $H^{+0}$ denotes the limit $h_\mathrm{N} \to \infty$ and $h_\mathrm{S} = 0$.

The defect changing and creation operators can be realized by setting $h_\mathrm{N} \neq h_\mathrm{S}$. By taking the limit $h_\mathrm{N} \to +\infty$ and $h_\mathrm{S} = 0$, we obtain the Hamiltonian $H^{+0}$. By the conformal mapping, this corresponds to a configuration with a line defect on the positive half of the $x^0$-axis of $\mathbb{R}^3$, and defect creation operators lie at the endpoints $0$ and $\infty$. Similarly, by taking the limit $h_\mathrm{N} \to +\infty$ and $h_\mathrm{S} \to -\infty$, we obtain the Hamiltonian $H^{+-}$ we create a line defect on the positive half of the $x^0$-axis of $\mathbb{R}^3$, and a line defect with opposite sign on the negative half, and defect changing operators lie at the endpoints $0$ and $\infty$.

We solve the system numerically through exact diagonalization up to $N = 17$. We also calculate the $g$-function and the scaling dimensions $\Delta_0^{+0}$ and $\Delta_0^{+-}$ through DMRG up to $N = 44$. We note that the Hilbert space is a direct sum of eigenspaces of the angular momentum $L^z$ along the $z$-axis. In this paper, we only consider the defect and defect changing operators in the $L^z = 0$ sector. The other $L^z$ sectors can be analyzed analogously.

Table 2: The 10 lowest defect creation and changing operators in the $L^z = 0$ sector from state-operator correspondence, organized in terms of the conformal multiplets, calculated at $N = 17$. All the operators are parity-even. The finite-size corrections have not been extrapolated yet in this table.

| $\phi_i^{+0}$ | $\Delta_i^{+0}$ | $\phi_i^{+-}$ | $\Delta_i^{+-}$ |
|---|---|---|---|
| $\phi_0^{+0}$ | 0.104 | $\phi_0^{+-}$ | 0.785 |
| $\partial_0\phi_0^{+0}$ | 1.084 | $\partial_0\phi_0^{+-}$ | 1.763 |
| $\partial_0^2\phi_0^{+0}$ | 2.077 | $\partial_0^2\phi_0^{+-}$ | 2.723 |
| $\partial_0^3\phi_0^{+0}$ | 3.075 | $\partial_0^3\phi_0^{+-}$ | 3.674 |
| $\partial_0^4\phi_0^{+0}$ | 4.055 | $\partial_0^4\phi_0^{+-}$ | 4.605 |
| $\phi_1^{+0}$ | 2.186 | $\phi_1^{+-}$ | 3.167 |
| $\partial_0\phi_1^{+0}$ | 3.234 | $\partial_0\phi_1^{+-}$ | 4.202 |
| $\partial_0^2\phi_1^{+0}$ | 4.225 | $\partial_0^2\phi_1^{+-}$ | 5.152 |
| $\phi_2^{+0}$ | 3.446 | $\phi_2^{+-}$ | 4.367 |
| $\phi_3^{+0}$ | 3.737 | $\phi_3^{+-}$ | 4.629 |

## 5.2 Spectrum of defect creation and changing operators

Before considering the overlaps, we first extract the scaling dimensions of the defect changing and creation operators from state-operator correspondence

$$\Delta_i^{+0} = \frac{1}{\tilde{R}}(E_i^{+0} - \tfrac{1}{2}E_0^{00} - \tfrac{1}{2}E_0^{++}) + CN^{-(\Delta_1^{++}-1)/2},$$

$$\Delta_i^{+-} = \frac{1}{\tilde{R}}(E_i^{+-} - E_0^{++}) + CN^{-(\Delta_1^{++}-1)/2}, \tag{51}$$

where $E_i^{ab}$ is the energy of the state corresponding to the operator $\phi_i^{ab}$ in the sector $ab$, and $\tilde{R} = (E_T^{00} - E_0^{00})/3$ is the energy spacing calibrated by the stress tensor in the bulk Ising CFT. The energy $E_i^{aa}$ and $E_i^{bb}$ are subtracted to cancel the non-universal contribution to the ground state energy. At finite sizes, these energies receive corrections from the irrelevant primaries of the defect CFT. The lowest one has scaling dimension $\Delta_1^{++} = 1.63$ taken form Ref. [27].

The lowest spectra of the defect creation and changing operators are listed in Table 2 measured at size $N = 17$, and several results at different sizes are plotted in Figure 4. These

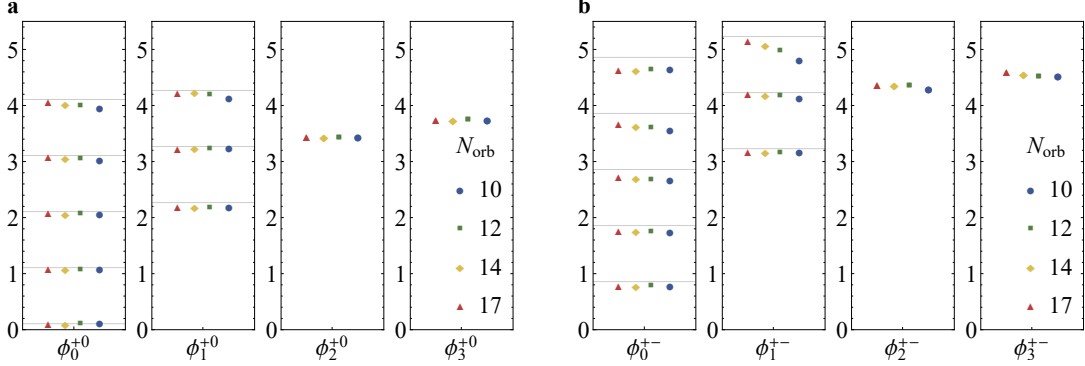

Figure 4: The scaling dimensions of defect creation (a) and changing (b) operators organized into different conformal multiplets at different system size $N$.

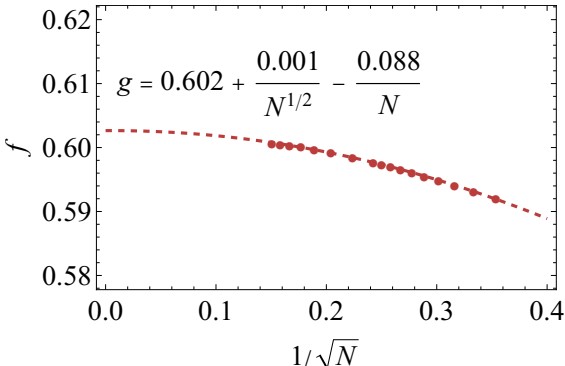

Figure 5: The finite-size extrapolation of the $g$-function. The data for $N \leq 17$ are obtained by ED and the data for $20 \leq N \leq 44$ are obtained by DMRG.

operators can be organized into multiplets with integer spacing. This is because the defect changing operators are organized into irreducible representations of SO(2, 1), and each primary operator $\phi_\alpha^{ab}$ is associated with a conformal tower $\partial_0^n \phi_\alpha^{ab}$ ($n = 0, 1, 2, \dots$) with scaling dimension $\Delta_\alpha^{ab} + n$. The integer spacing is a strong indication of the conformal symmetry of the defect.

## 5.3 The $g$-function

The first conformal data we extract from overlap is the $g$-function of the Ising CFT with magnetic defect. This can be derived from Eq. (40) by taking $a = +$. The finite-size correction is given by Eq. (47)

$$g(N) = g + C_1 N^{-1/2} + C_2 N^{-1}, \tag{52}$$

where the radius is taken as $R = \sqrt{N}$ and $C_1$ and $C_2$ are constants to be fitted (Figure 5). After the extrapolation, we obtain

$$g = 0.602(2).$$

The error bar is estimated from finite-size extrapolation. Throughout this paper, we take the error bar from finite-size scaling to be the difference between the extrapolated value and the data of the largest possible system size. We also need to note that the error bars given are not strict but only an estimation. We can see that the finite size result is already close to the extrapolated value. The finite-size scaling is thus not indispensable but only serves to further improve the numerical accuracy. We also compare this result to the perturbation calculations. To the first order in $(4-\epsilon)$-expansion, $g = \exp(-\epsilon(N+8)/16) + \mathcal{O}(\epsilon^2) = 0.57 + \mathcal{O}(\epsilon^2)$ [26]; To the first order in large-$N$ expansion, $g = \exp(-0.153673N) + \mathcal{O}(N^0) = 0.857 + \mathcal{O}(N^0)$ [26].

## 5.4 Scaling dimensions of defect changing operators from overlap

The next information we can obtain is the scaling dimensions of defect creation and changing operator. Using the overlap method, one way is to extract them from the ratio (43). By taking $(a, b) = (+, 0), (0, +)$ and $(+, -)$, we get[9]

$$2^{-2\Delta_0^{+0}} = A_{00}^{++00}/A_{00}^{+000} A_{00}^{+++0} + C_1 N^{-1/2} + C_2 N^{-1},$$
$$2^{-2\Delta_0^{+-}} = A_{00}^{++--}/(A_{00}^{+++-})^2 + C_2 N^{-1}, \tag{53}$$

---

[9]Note that when taking the overlap $\langle +b| -b \rangle$, an $n^x(+\hat{z})$ is acted implicitly on the interface. Otherwise, the overlap will be strictly zero.

Table 3: The scaling dimension of defect creation operators $\Delta_0^{+0}$, $\Delta_1^{+0}$ and defect changing operator $\Delta_0^{+-}$ obtained from different methods. The finite size results are plotted in Figure 6. The error bars are estimated from finite-size extrapolation.

| Method | $\Delta_0^{+0}$ | $\Delta_1^{+0}$ | $\Delta_0^{+-}$ |
|---|---|---|---|
| From overlap ratio (53) | 0.107(3) | | 0.83(4) |
| From overlap scaling (47) | 0.109(4) | 2.23(19) | 0.85(3) |
| From energy spectrum (51) | 0.107(3) | 2.25(7) | 0.85(5) |
| Maximal discrepancy | 0.7% | 1.5% | 5.5% |

where we fit over $C_{1,2}$ to extrapolate the ratio in the thermodynamic limit(Figure 6a,c, blue squares). The other way is to fit from the size dependence of the overlaps $A_{00}^{+000}$ and $A_{00}^{+++-}$ with Eq. (47) (Figure 6d,f). The scaling dimension of the second defect creation primary $\phi_1^{+0}$ can also be extracted similarly from the overlap $A_{10}^{+000}$ (Figure 6e). We can compare these results with the scaling dimensions extracted from state-operator correspondence (Figure 6a–c, red circles). The values obtained from different methods are listed in Table 3. The results are all consistent within a discrepancy of 5%.

## 5.5 OPE coefficients

We then turn to the OPE coefficients between the defect creation, changing operators, and the bulk and defect primary operators.

The first OPE coefficient we calculate is $C_{000}^{+0-}$ between two defect creation operators $\phi_0^{\pm 0}$ and a defect changing operator $\phi_0^{+-}$. by taking $a, b, c = +, 0, -$ and $\alpha = \gamma = 0$ in Eq. (44) we

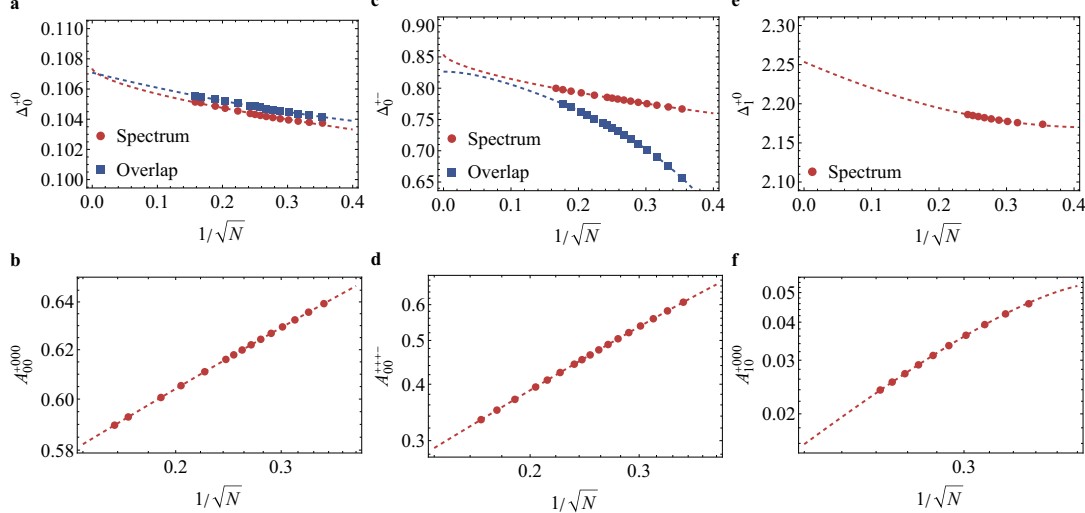

Figure 6: (a–c) The scaling dimensions of the defect creation operators $\Delta_0^{+0}$ (a), $\Delta_1^{+0}$ (b) and defect changing operator $\Delta_0^{+-}$ (c) extrapolated through the state-operator correspondence (51) and the overlap ratios (43); (d–f) the dependence of the overlaps $a_{00}^{+000}$ (d), $a_{10}^{+000}$ (e) and $a_{00}^{+++-}$ (f) on $1/\sqrt{n}$, where the power in the scaling gives the scaling dimensions $\delta_0^{+0}, \delta_1^{+0}$ and $\delta_0^{+-}$ respectively through eq. (47). the data for $n \leq 17$ are obtained by ed and the data for $20 \leq N \leq 40$ are obtained by DMRG.

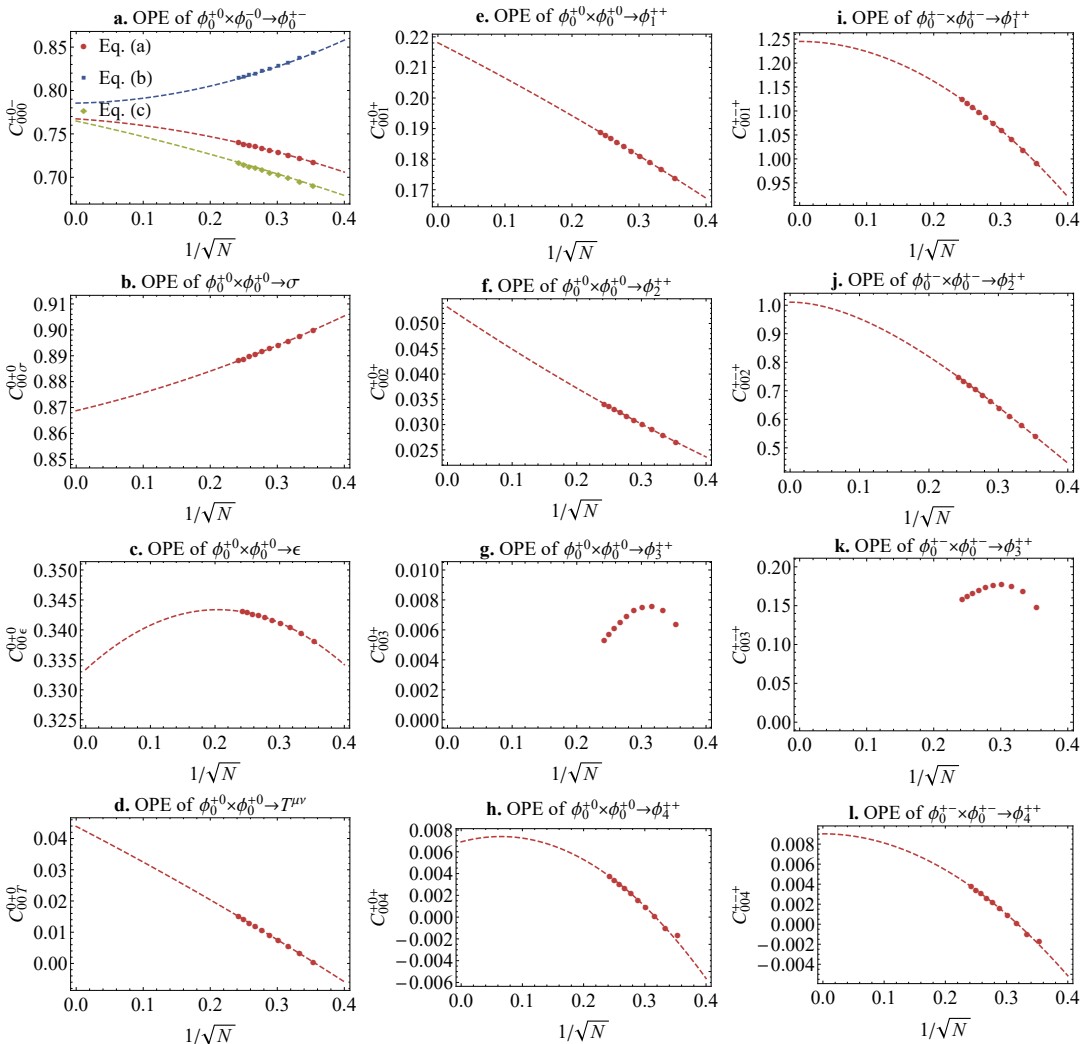

Figure 7: The finite size extrapolation of OPE coefficients (a) of defect changing operator $\phi_0^{+-}$, $\phi_0^{+-}$ and defect operator $\phi_\alpha^{++}$ obtained through Eq. (54), OPE coefficients $C_{00\alpha}^{0+0}$ (b–d) of two defect creation operator $\phi_0^{+0}$ and bulk operator $\phi_\alpha^{00}$, and OPE coefficients $C_{00\alpha}^{+0+}$ (e–h), $C_{00\alpha}^{+-+}$ (i–l) of two defect creation operators $\phi_0^{+0}$ or two defect changing operators $\phi_0^{+-}$ with defect operator $\phi_\alpha^{++}$ obtained through Eq. (55). The extrapolated results are listed in Tables 4 and 5.

obtain

$$C_{000}^{+0-} = A_{00}^{+-+0}/A_{00}^{+000}, \tag{54a}$$

$$C_{000}^{+0-} = (1/\sqrt{g})A_{00}^{+0-0}/A_{00}^{+++-}, \tag{54b}$$

$$C_{000}^{+0-} = (1/2^{\Delta_0^{+-}}\sqrt{g})A_{00}^{+-00}/A_{00}^{++00}. \tag{54c}$$

After a finite size extrapolation (where the finite-size corrections come from the first and second-order derivative descendants of the defect creation or changing operators), we obtain from these three expressions

$$C_{000}^{+0-} = 0.767(27), 0.785(28), 0.77(5).$$

These results are consistent within an error of 2.5%.

We then consider the OPE coeffients $C_{00\alpha}^{0+0}$ between two defect creation operators $\phi_0^{+0}$ and a bulk primary $\phi_\alpha^{00}$ by setting $a = c = 0, b = +$ in Eq. (44), the OPE coeffients $C_{00\alpha}^{0+0}$ between two defect creation operators $\phi_0^{+0}$ and a defect primary $\phi_\alpha^{++}$ by setting $a = c = +, b = 0$ in Eq. (44), and the OPE coeffients $C_{00\alpha}^{+-+}$ between two defect changing operators $\phi_0^{+-}$ and a defect primary $\phi_\alpha^{++}$ by setting $a = c = +, b = -$ in Eq. (44).

$$C_{00\alpha}^{0+0} = A_{0\alpha}^{+000}/A_{00}^{+000} + C_1 N^{-1/2} + C_2 N^{-1},$$
$$C_{00\alpha}^{+0+} = A_{0\alpha}^{+0++}/A_{00}^{+0++} + C_1 N^{-1/2} + C_2 N^{-1},$$
$$C_{00\alpha}^{+-+} = A_{0\alpha}^{+-++}/A_{00}^{+-++} + C_2 N^{-1}, \tag{55}$$

where the finite size correction to the first two overlap ratios comes from the first and second order descendant of $\phi_0^{+0}$ and the finite size correction to the last overlap ratio comes from the first and second order descendant of $\phi_0^{+-}$. The finite size analysis is plotted in Figure 7 and the extrapolated results are listed in Tables 4 and 5. Note that due to the non-monotonic behavior with system size, $C_{003}^{+0+}$ and $C_{003}^{+-+}$ are hard to extrapolate. However, from the finite size results, we can see that $C_{003}^{+0+}$, as well as $C_{004}^{+0+}$, is at least one magnitude smaller than other OPE coefficients.

## 5.6 Scaling dimensions of bulk and defect primaries

From the overlaps, we can also obtain the scaling dimensions of the primaries of the bulk Ising CFT and the defect CFT. The scaling dimensions can also be obtained by the energy spectrum. Therefore, this section serves as a consistency check of the two methods. By substituting $(a, b) = (0, +), (+, 0)$ and $(+, -)$ into Eq. (45), we obtain

$$2^{\Delta_\alpha^{00}} = (1/C_{00\alpha}^{0+0})A_{0\alpha}^{++00}/A_{00}^{++00} + C_1 N^{-1/2} + C_2 N^{-1}, \tag{56}$$

$$2^{\Delta_\alpha^{++}} = (1/C_{00\alpha}^{+0+})A_{\alpha0}^{++00}/A_{00}^{++00} + C_1 N^{-1/2} + C_2 N^{-1}, \tag{57}$$

$$2^{\Delta_\alpha^{++}} = (1/C_{00\alpha}^{+-+})A_{\alpha0}^{++--}/A_{00}^{++--} + C_2 N^{-1}. \tag{58}$$

The finite size analysis is plotted in Figure 8 and the extrapolated results are listed in Tables 4 and 5. We compare the obtained scaling dimensions with the results of the numerical bootstrap for the bulk operators [62] and the results from the state-operator correspondence [27] for the defect operators, and they are consistent within a discrepancy of 1.5% .[10]

Table 4: The OPE coefficients $C_{00\alpha}^{0+0}$ of two defect creation operators $\phi_0^{+0}, \phi_0^{+0}$ with bulk operator $\phi_\alpha^{00}$ obtained through Eq. (55), and the scaling dimensions $\Delta_\alpha^{00}$ of bulk operators obtained through Eq. (56) compared with the result from numerical bootstrap $\Delta_\alpha^{00}$ [62]. The slash denotes that we are not able to extrapolate the data from current system sizes. The finite size results are plotted in Figs. 7 and 8. The error bars are estimated from finite-size extrapolation.

| $\phi_\alpha^{00}$ | $C_{00\alpha}^{0+0}$ | $\Delta_\alpha^{00}$, Eq. (56) | $\Delta_\alpha^{00}$, Ref. [62] | Error |
|---|---|---|---|---|
| $\sigma$ | 0.869(19) | 0.520(5) | 0.518 | 0.4% |
| $\epsilon$ | 0.3334(9) | 1.431(23) | 1.413 | 1.3% |
| $T^{\mu\nu}$ | 0.044(28) | / | | |
| $\epsilon'$ | 0.003(7) | / | | |

---

[10]Note that in Ref. [27], the operators $\phi_{1,2,3}^{++}$ in our paper are referred to as $\hat\phi, \hat\phi', \hat\phi''$.

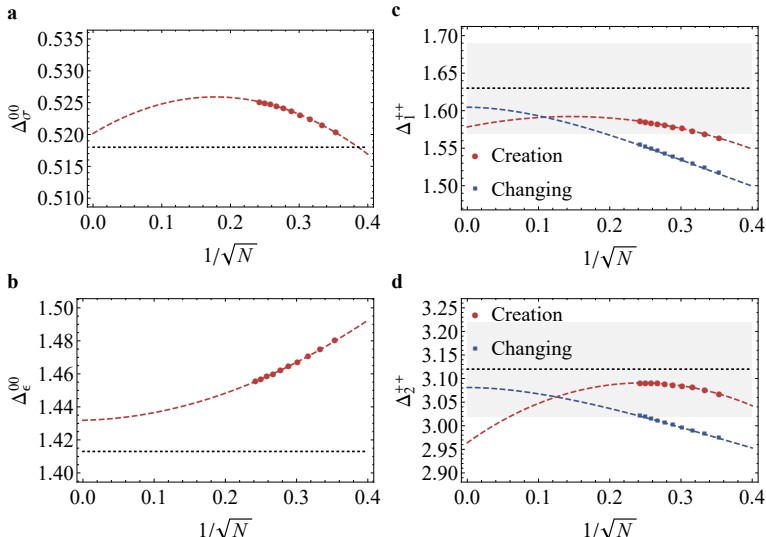

Figure 8: The finite size extrapolation the scaling dimensions $\Delta_\alpha^{00}$ (a,b) of bulk operators and $\Delta_\alpha^{++}$ (c,d) of defect operators obtained through Eq. (56). The dotted line denotes the results from numerical bootstrap [62] or the state-operator correspondence on fuzzy sphere [27] and the shaded region denotes the range of error bar. The extrapolated results are listed in Tables 4 and 5.

Table 5: The OPE coefficients $C_{00\alpha}^{+0+}$ and $C_{00\alpha}^{+-+}$ of two defect creation operators $\phi_0^{+0}$ or two defect changing operators $\phi_0^{+-}$ with defect operator $\phi_\alpha^{++}$ obtained through Eq. (55), and the scaling dimensions $\Delta_\alpha^{++}$ of defect operators obtained through Eqs. (57) and (58) compared with the result from state operator correspondence $\Delta_\alpha^{00}$ [27]. The slash denotes that we are not able to extrapolate the data from current system sizes. The finite size results are plotted in Figs. 7 and 8. The error bars are estimated from finite-size extrapolation.

| $\phi_\alpha^{++}$ | $C_{00\alpha}^{+0+}$ | $C_{00\alpha}^{+-+}$ | | $\Delta_\alpha^{++}$ | | Error |
|---|---|---|---|---|---|---|
| | | | Eq. (57) | Eq. (58) | Ref. [27] | |
| $\phi_1^{++}$ | 0.22(3) | 1.25(12) | 1.58(2) | 1.61(5) | 1.63(6) | 1.5% |
| $\phi_2^{++}$ | 0.053(19) | 1.01(26) | 2.96(2) | 3.08(6) | 3.12(13) | 1.3% |
| $\phi_3^{++}$ | / | / | / | / | | |
| $\phi_4^{++}$ | 0.007(3) | 0.009(5) | / | 4.4(3) | / | / |

Table 6: The ratio of overlaps $A^{aaad}_{0i;0n}/A^{aaad}_{0i}$ for descendant operators $\partial_0 \phi^{+0}_0$, $\partial^2_0 \phi^{+0}_0$ and $\partial^2_0 \phi^{+-}_0$ extrapolated to thermodynamic limit compared with the theoretical value calculated from Eq. (59). The finite size results are plotted in Figure 9. The error bars are estimated from finite-size extrapolation.

| Operator | Overlap | Numerical | Eq. (59) | Descrepancy |
|---|---|---|---|---|
| $\partial_0 \phi^{+0}_0$ | $A^{+000}_{00;10}/A^{+000}_{00}$ | 0.46(4) | 0.463 | 1.7% |
| $\partial^2_0 \phi^{+0}_0$ | $A^{+000}_{00;20}/A^{+000}_{00}$ | 0.36(8) | 0.361 | 0.8% |
| $\partial^2_0 \phi^{+-}_0$ | $A^{+++-}_{00;02}/A^{+++-}_{00}$ | 1.5(4) | 1.497 | 0.3% |

## 5.7 Descendant operators

We can also numerically verify that the operators with integer spacing are in the same conformal family using the overlap methods. The ratio between $A^{aaad}_{0i;0n} = \langle \partial^n_\tau \phi^{ad}_i | \mathbb{I}^{aa} \rangle$ and $A^{aaad}_{0i}$ is uniquely fixed by the scaling dimension $\Delta^{ad}_i$

$$\frac{A^{aaad}_{0i;0n}}{A^{aaad}_{00}} = \sqrt{\frac{\Gamma(n+2\Delta^{ad}_i)}{n!\Gamma(2\Delta^{ad}_i)}} \, . \tag{59}$$

We calculate the overlap $A^{+++0}_{00;01}, A^{+++0}_{00;02}, A^{+++-}_{00;02}$ for first and second derivative of $\phi^{+0}_0$ and the second derivative of $\phi^{+-}_0$. The extrapolated result from the overlap and the theoretical value calculated from Eq. (59) agrees within a discrepancy of 2%. The results are listed in Table 6 and plotted in Figure 9.

# 6 The stability of Ising SSB defects

We now use the defect conformal data presented above to address an intriguing question — the stability of spontaneous symmetry breaking (SSB) on the defect line. As is well known, for a pure one-dimensional local system (such as a classical Ising chain), SSB is not a stable phase due to the negative free energy cost of the domain wall. This is no longer obvious when the one-dimensional system is inserted into a higher-dimensional bulk CFT as a defect, as the defect is strongly interacting with the bulk matter.

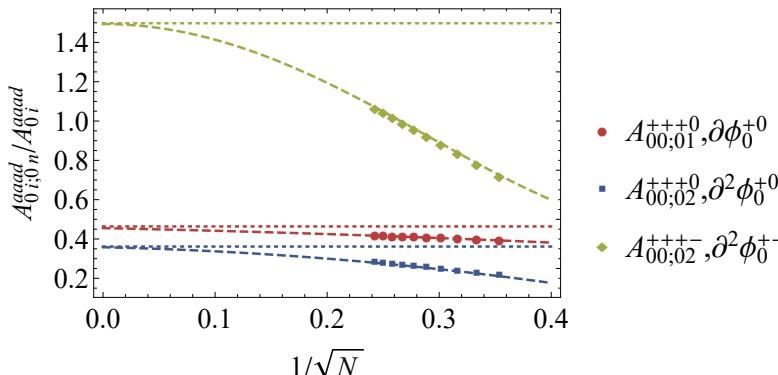

Figure 9: The ratio of overlaps $A^{aaad}_{0i;0n}/A^{aaad}_{0i}$ where the dotted lines denote the theoretical value calculated from Eq. (59). The extrapolated results are listed in Table 6.

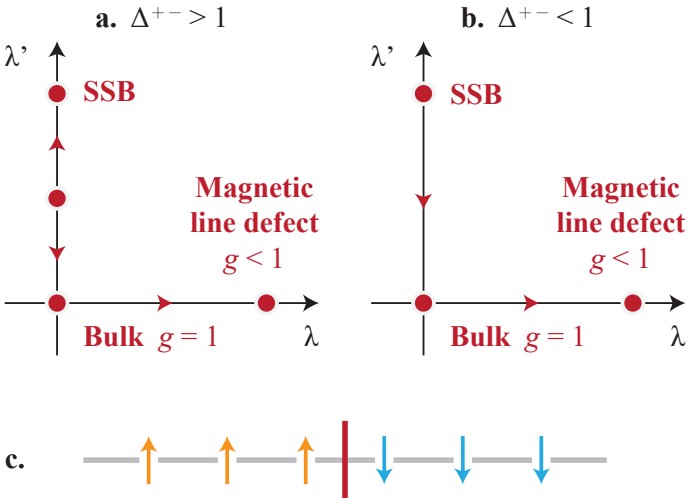

Figure 10: (a–b) The RG flow between bulk, magnetic line defect and SSB line defect in the cases of (a) $\Delta_0^{+-} > 1$ and (b) $\Delta_0^{+-} < 1$ and $g > 0.5$, where $\lambda$ controls a perturbation on the line defect that breaks $\mathbb{Z}_2$ symmetry, and $\lambda'$ controls a perturbation on the line defect that preserves $\mathbb{Z}_2$ symmetry. (c) An illustration of a domain wall on a 1D lattice spin chain.

The result of spontaneous symmetry breaking is a superpostion of simple line defects. For the $\mathbb{Z}_2$ symmetry, an SSB defect is in the superposition of $+$ and $-$. Unlike the simple defect which has a unique identity operator $\phi_0^{aa}$ with $\Delta_0^{aa} = 0$ and $\partial\phi_0^{aa} = 0$, the non-simple line defects are characterized by the presence of multiple local operators of scaling dimension 0: identity operators for each summand line defect that act on mutually orthogonal super-selection sectors.

One criterion to determine the stability of SSB defect is the relevance of the lowest defect changing operator $\phi_0^{+-}$. As mentioned above, this operator corresponds to creating a domain wall on the defect line (Figure 10c). This operator does not violate the $\mathbb{Z}_2$ symmetry of the SSB defect and could be therefore a perturbation to the dCFT. If $\Delta_0^{+-} > 1$, the domain wall operator is irrelevant, and the SSB defect is thus stable against the perturbation (Figure 10a). If $\Delta_0^{+-} < 1$, the domain wall operator is relevant, which leads to domain wall proliferation and drives the defect to a more stable disordered phase (Figure 10b).

Another evidence of the instability of the SSB defect comes from the value of $g$-function. The $g$-function we calculate is within the range $0.5 < g < 1$. The $g$-theorem, asserting the monotonicity of $g$-function of the RG scale, first is consistent with our computed g-function $g = 0.602(2) < g_{\text{bulk}} = 1$. On the other hand, the SSB defect, as a superposition of two copies of magnetic line defect, would have twice the $g$-function of magnetic line defect $g_{\text{SSB}} = 2g > 1$, so this is consistent with SSB defect is unstable and will likely flow to the transparent defect (bulk CFT).

# 7 Discussion

In this work, we have used the wavefunction overlap on the fuzzy sphere to study the 3D Ising CFT with a magnetic line defect. We have computed the $g$-function, the scaling dimensions of defect creation operators and defect changing operators, and their OPE coefficients. One feature of our method is that each piece of conformal data may be computed using many different

overlaps. We have checked that all the numerical results are consistent with each other. The scaling dimensions of bulk and defect operators are also extracted from the overlap, which agrees with previous results based on the energy spectrum. The computation demonstrates again that the fuzzy sphere regularization is close to the conformal fixed point even at small sizes. The conformal data we computed have provided evidence for the instability of the Ising SSB defect, including $\Delta_0^{+-} < 1$ and $g > 0.5$.

The method of extracting conformal data from overlaps can be applied to other various defect CFTs. It is not known if other conformal line defects may exist in the 3d Ising CFT. A small variant that can be defined in the 3d Ising model is the monodromy defect [20]. In the Ising CFT, this is formulated by changing the sign of interaction on an infinite half-plane attached to the defect line. The half-plane itself is topological, but the line defect attached to it is not.

Besides defects in Ising CFT, this method can also be applied to defects in other CFTs that can be realized in the fuzzy sphere, such as O($N$) Wilson-Fisher CFT [74] and the SO(5) Wess-Zumino-Witten model [73]. These models can potentially exhibit a much richer structure of defect fixed points, such as spin impurities in the O(3) Wilson-Fisher CFT [38]. One can also consider the gauge theories and study the Wilson line operators.

In addition to line defects, the overlap method can also help the study of defect CFTs with various other geometries. If we place the line defects at points with angular distance $\theta \neq \pi$ instead of the north and south poles, we can realize a "cusp" configuration involving two infinite half lines joining at the same point. The universal part of the ground state energy $E(\theta)$ corresponds to the scaling dimension of the cusp defect.

Another potential generalization of this work is to go beyond line defects. Turning on a perturbation on the equator of the fuzzy sphere, we can realize 2D defects, i.e., boundaries and interfaces. Alternatively, the overlap between the bulk wavefunction at the critical point and the bulk wavefunction away from criticality creates a boundary at $\tau = 0$ between the CFT and polarised states. The monotonic function controlling the RG flow of 2d defects, known as the $B$-function, can be extracted in this manner [78].

# Acknowledgments

We would like to thank Zohar Komargodski, Ryan Lanzetta, Marco Meineri, and Max Metlitski for illuminating discussions. Part of the numerical calculations are done using the packages FuzzifiED and ITensor.

**Funding information** Z.Z. acknowledges support from the Natural Sciences and Engineering Research Council of Canada (NSERC) through Discovery Grants. Research at Perimeter Institute is supported in part by the Government of Canada through the Department of Innovation, Science and Industry Canada and by the Province of Ontario through the Ministry of Colleges and Universities.

# A Descendant states

The defect operator and defect changing operators are organized into irreps of SO(2,1). Each primary operator $\phi_\alpha^{ab}$ is associated with a conformal tower $\partial_\tau^n \phi_\alpha^{ab}$, where $n = 0, 1, 2, \cdots$, with the scaling dimension $\Delta_\alpha^{ab} + n$. We are interested in the overlap

$$A_{\alpha\beta;mn}^{abcd} = \langle \partial_\tau^n \phi_\beta^{cd} | \partial_\tau^m \phi_\alpha^{ab} \rangle. \tag{A.1}$$

The derivation that leads to Eq. (39) applied to descendant states gives

$$A_{\alpha\beta;mn}^{abcd} = M_0^{ca} M_0^{bd} \left(\frac{1}{R}\right)^{\Delta_0^{ca}+\Delta_0^{bd}} e^{-\delta_{ab}\log g_a/2} e^{-\delta_{cd}\log g_c/2} \langle \partial_\tau^n \phi_\beta^{cd} | \phi_0^{ca}(-1)\phi_0^{bd}(1)|\partial_\tau^m \phi_\alpha^{ab}\rangle.$$
(A.2)

The state-operator correspondence of descendant states is slightly different from the primary state

$$|\partial_\tau^m \phi_\alpha^{ab}\rangle = C(m,\Delta_\alpha^{ab})\partial_\tau^m \phi_\alpha^{ab}(0)|\mathbb{I}\rangle,$$
(A.3)

$$\langle \partial_\tau^n \phi_\beta^{cd}| = C(n,\Delta_\beta^{cd}) \lim_{\tau\to\infty} \langle I|(\tau^2\partial_\tau)^n(\tau^{2\Delta_\beta^{cd}}\phi_\beta^{dc}(\tau)),$$
(A.4)

where

$$C(m,\Delta)^{-2} = m!\frac{\Gamma(m+2\Delta)}{\Gamma(2\Delta)},$$
(A.5)

is a normalization constant such that the normalization $\langle \partial_\tau^n \phi_\beta^{ab}|\partial_\tau^m \phi_\alpha^{ab}\rangle = \delta_{mn}\delta_{\alpha\beta}$ holds. Note that the definition of the bra state, Eq. (A.4), essentially follows from Eq. (A.3) by

$$\begin{aligned}
\partial_\tau^n \phi_\beta^{cd}(0)^\dagger &= \lim_{\tau'\to 0} \partial_{\tau'}^n \phi_\beta^{cd}(\tau')^\dagger \\
&= \lim_{\tau'\to 0} \partial_{\tau'}^n \left(\frac{1}{\tau'^{2\Delta_\beta^{cd}}}\phi^{dc}\left(\frac{1}{\tau'}\right)\right) \\
&= \lim_{\tau\to\infty}(\tau^2\partial_\tau)^n(\tau^{2\Delta_\beta^{cd}}\phi_\beta^{dc}(\tau)),
\end{aligned}$$
(A.6)

where in the last line we have changed the variable from $\tau'$ to $\tau = 1/\tau'$. As in the main text, we would like to simplify the four-point correlation function Eq. (A.2) to a two or three-point correlation function by setting some operators to identity. The simplest example is to set $a = b = c \neq d, m = 0, \alpha = \beta = 0$. This reduces the four-point correlation function to a two-point correlation function, which results in

$$\frac{A_{00;0n}^{aaad}}{A_{00}^{aaad}} = \langle \partial_\tau^n \phi_0^{ad}|\phi_0^{ad}(1)|\mathbb{I}^{aa}\rangle,$$
(A.7)

where $A_{00}^{aaad}$ equals exactly the first line of Eq. (A.2). Now we can substitute Eq. (A.4) into the correlator and use $\langle \phi_0^{da}(\tau)\phi_0^{ad}(1)\rangle = (\tau-1)^{-2\Delta_0^{ad}}$. The final result is given by [66]

$$\frac{A_{00;0n}^{aaad}}{A_{00}^{aaad}} = \sqrt{\frac{\Gamma(n+2\Delta_0^{ad})}{n!\Gamma(2\Delta_0^{ad})}}.$$
(A.8)

Exploiting the fact that the defect junction contains all symmetry-allowed defect changing operators, this formula can be readily generalized to other primary states (if $\phi_\beta^{ad}$ is symmetry-allowed at the junction),

$$\frac{A_{0\beta;0n}^{aaad}}{A_{0\beta}^{aaad}} = \sqrt{\frac{\Gamma(n+2\Delta_\beta^{ad})}{n!\Gamma(2\Delta_\beta^{ad})}}.$$
(A.9)

Note that Eq. (A.9) also implies an efficient way to check if one state in the defect changing sector is a descendant state of a given primary. This will be useful if a descendant state has a scaling dimension close to some other primary state such that it cannot be clearly distinguished based solely on energy levels. We have numerically checked that Eq. (A.9) holds for the 3D Ising model for $a = +, b = 0$ and $a = +, b = -$ up to $n \leq 3$.

Finally, we comment that similar derivation can be applied to other overlaps involving the three-point correlation function of descendant operators, which are in turn completely fixed by the OPE coefficients and conformal invariance.

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
