# Peer review of "The $g$-function and Defect Changing Operators from Wavefunction Overlap on a Fuzzy Sphere"

_SciPost Physics, doi:SciPost Phys. 17, 021 (2024)_

## Round 2 · Referee Report · Anonymous (Referee 1) · 2024-4-3

Report

In this paper, the fuzzy sphere regularization technique is applied to the magnetic line defect in the 3d Ising CFT. The technique consists in a specific realization of the statistical universality class of interest, by means of interacting fermions in a uniform magnetic field on a sphere. The inverse of the magnitude of the magnetic field provides the regulator. A line defect can be realized by modifying the Hamiltonian at the two poles of the sphere. The magnetic line defect, in particular, is a $\mathbb{Z}_2$ breaking deformation, which can also be realized, for instance, as the (IR limit of) a magnetic field localized on a line, in the 3d classical Ising model.

The authors extract the scaling dimensions of various low lying operators on the line, as well as of operators that interpolate between the different sectors of the defect theory (+ line, - line, no line). The fixed point value of the RG monotonic $g$ function is computed as well. The paper provides the first determination of most of these quantities. Some of these estimates also have qualitative consequences, most notably excluding the possibility of generating the line defect from a localized $\mathbb{Z}_2$ even deformation, via spontaneous symmetry breaking.

The paper is clearly written, well organized, and definitely meets the standards for publication on SciPost. I will point out a typo and make a few minor observations below: the authors can optionally address them in a revised version.

  1. One of the main results in the paper is the no-SSB statement alluded to above. The result follows from the fact that the domain wall creation operator is relevant. Yet, the explanation on why one fact implies the other, and in fact any discussion of SSB on line defects at all is relegated to a few lines on page 4, page 23 and a few footnotes. There is also no reference to the literature, which could have guided the non-expert reader in lack of a coherent explanation. While intuitive, the precise definition of a SSB defect and the mechanism which destabilizes it deserve a clearer treatment, rather than a few scattered comments.

  2. The ability to measure the $g$ quantity follows from the fact that the defect is unambiguously normalized. This is visible in eq. 4 from the absence of non-universal constants associated to the operators inserted at 0 and infinity. This requirement appears to be slightly different from the normalization of the identity as expressed in footnote 8. Changing the Hamiltonian (30) by addition of a $a$ and $b$-dependent constant would in particular modify the value of $g$. The authors might want to explain why such a constant cannot be generated by the RG flow relating their UV realization to the defect CFT. For instance, such ambiguity would be present, at least naively, starting with order $\lambda^2$ in conformal perturbation theory, if one realizes the defect by explicit integration of $\sigma$ in the CFT path integral.

  3. Right below eq. (30), $\mathcal{O}_\beta$ should read $\mathcal{O}_b$.

  4. At the end of page 19, the authors note that the finite size behavior of the OPE coefficients involving heavy defect operators is harder to extrapolate due to non-monotonicity. Why is this the case? Are subleading finite-size corrections expected to be more important in this case? The authors might want to add some further explanation, if available.

Requested changes

Optional clarifications as detailed in the report

  • validity: top
  • significance: high
  • originality: high
  • clarity: top
  • formatting: good
  • grammar: perfect

Author:  Zheng Zhou  on 2024-06-18  [id 4574]

(in reply to Report 1 on 2024-04-03)
Category:
remark
answer to question

We thank the referee for reviewing our paper and for his/her recommendation of our paper and the detailed comments. In the following, we will give a point-by-point reply to the comments by the referee.

One of the main results in the paper is the no-SSB statement alluded to above. The result follows from the fact that the domain wall creation operator is relevant. Yet, the explanation on why one fact implies the other, and in fact any discussion of SSB on line defects at all is relegated to a few lines on page 4, page 23 and a few footnotes. There is also no reference to the literature, which could have guided the non-expert reader in lack of a coherent explanation. While intuitive, the precise definition of a SSB defect and the mechanism which destabilizes it deserve a clearer treatment, rather than a few scattered comments.

We thank the referee for the helpful suggestion. In the revised manuscript, we have devoted a whole section to discuss the SSB defects.

The ability to measure the $g$ quantity follows from the fact that the defect is unambiguously normalized. This is visible in eq. 4 from the absence of non-universal constants associated to the operators inserted at 0 and infinity. This requirement appears to be slightly different from the normalization of the identity as expressed in footnote 8. Changing the Hamiltonian (30) by addition of a $a$ and $b$-dependent constant would in particular modify the value of $g$. The authors might want to explain why such a constant cannot be generated by the RG flow relating their UV realization to the defect CFT. For instance, such ambiguity would be present, at least naively, starting with order $\lambda^2$ in conformal perturbation theory, if one realizes the defect by explicit integration of $\sigma$ in the CFT path integral.

The higher CFT operators that overlap with the UV realisations $\mathcal{O}_{a,b}$ has scaling dimensions $>1$ and is irrelevant under the RG flow. The contribution, proportional to $\lambda^2$ in the conformal perturbation theory, scales to 0 in the thermodynamic limit. Different forms of the Hamiltonian might have the value of $g$ at finite size, but under the finite size scaling, they will extrapolate to the same universal value in the thermodynamic limit.

Right below eq. (30), $\mathcal{O}_\beta$ should read $\mathcal{O}_b$

We thank the reviewer for the careful reading. We have fixed the mistake in the revised version of the manuscript.

At the end of page 19, the authors note that the finite size behavior of the OPE coefficients involving heavy defect operators is harder to extrapolate due to non-monotonicity. Why is this the case? Are subleading finite-size corrections expected to be more important in this case? The authors might want to add some further explanation, if available.

The difficulty of finite size extrapolation just happens to the operator $\phi_3^{++}$. We found that the quadratic fitting according to Eq. (55) does not quite capture the dependence of $C_{003}^{+0+}$ on $1/\sqrt{N}$, and fittings involving higher subleading contributions would lead to instability that the fitting error is much larger than the value itself, so we choose not to present the fitting in the manuscript. This observation heavily depend on the UV realisation of the dCFT and so far we do not have a deeper explanation.

We hope that the referee would agree that this manuscript is suitable for publication with the revisions made.

---

## Round 2 · Referee Report · Anonymous (Referee 2) · 2024-6-2

Strengths

well organized and clear

Report

The authors study defect CFTs in conformal field theory using overlap wavefuctions and they are able to extract many information about the defect, incuding the g-funcion, some scaling dimensions and OPEs. They apply the techniques to the magnetic defects of the 3d Ising CFT (using the fuzzy sphere regularization) and they are able to extract numerical results. The authors give the first determination of many of these quantities. The paper is timely and well-written. It also contains a nice summary section at the very beginning. In my opinion the paper meets the SciPost criteria and I recommend it for publication.

Recommendation

Publish (surpasses expectations and criteria for this Journal; among top 10%)

---

## Round 3 · List of Changes



---

## Editorial Decision

published